# Antipsychotic drugs selectively decorrelate long-range interactions in deep cortical layers

**Matthias Heindorf[1], Georg B Keller[1,2]***

[1]Friedrich Miescher Institute for Biomedical Research, Basel, Switzerland; [2]Faculty of Science, University of Basel, Basel, Switzerland

**Abstract** Psychosis is characterized by a diminished ability of the brain to distinguish externally driven activity patterns from self-generated activity patterns. Antipsychotic drugs are a class of small molecules with relatively broad binding affinity for a variety of neuromodulator receptors that, in humans, can prevent or ameliorate psychosis. How these drugs influence the function of cortical circuits, and in particular their ability to distinguish between externally and self-generated activity patterns, is still largely unclear. To have experimental control over self-generated sensory feedback, we used a virtual reality environment in which the coupling between movement and visual feedback can be altered. We then used widefield calcium imaging to determine the cell type-specific functional effects of antipsychotic drugs in mouse dorsal cortex under different conditions of visuomotor coupling. By comparing cell type-specific activation patterns between locomotion onsets that were experimentally coupled to self-generated visual feedback and locomotion onsets that were not coupled, we show that deep cortical layers were differentially activated in these two conditions. We then show that the antipsychotic drug clozapine disrupted visuomotor integration at locomotion onsets also primarily in deep cortical layers. Given that one of the key components of visuomotor integration in cortex is long-range cortico-cortical connections, we tested whether the effect of clozapine was detectable in the correlation structure of activity patterns across dorsal cortex. We found that clozapine as well as two other antipsychotic drugs, aripiprazole and haloperidol, resulted in a strong reduction in correlations of layer 5 activity between cortical areas and impaired the spread of visuomotor prediction errors generated in visual cortex. Our results are consistent with the interpretation that a major functional effect of antipsychotic drugs is a selective alteration of long-range layer 5-mediated communication.

**\*For correspondence:**
georg.keller@fmi.ch

## eLife assessment

This **important** study uses calcium imaging in mice to advance our understanding of the effect of antipsychotic drugs on neural functioning. The evidence supporting the conclusions is **convincing**, and this work will be of interest to neuroscientists working on visual processing and psychosis researchers.

## Introduction

Under normal conditions, self-generated movements are coupled to sensory feedback in different modalities. Locomotion, for example, is coupled to proprioceptive feedback from muscle movements, somatosensory and auditory feedback from touching the ground, vestibular feedback from translational acceleration, and visual feedback in the form of expanding visual flow. This coupling is key to our brain's ability to learn internal models of the world (*Jordan and Rumelhart, 1992*). Internal

models are the transformations between cortical areas and can be used to predict the sensory consequences of movement. Hallucinations and delusions can be explained as a misconfiguration of these internal models: inaccurate predictions of the sensory consequences of movements can result in self-generated activity that is not correctly distinguished from externally driven activity (*Fletcher and Frith, 2009*; *Frith, 2005*).

Self-generated movements result in broad activation of most regions of the brain (*Musall et al., 2019*; *Stringer et al., 2019*), including some primary sensory areas like visual cortex (*Keller et al., 2012*; *Saleem et al., 2013*) where they have been shown to depend on visual context (*Pakan et al., 2016*). The framework of predictive processing postulates that in cortex, these movement-related signals are internal model-based predictions of the sensory consequences of movement that are compared to externally generated bottom-up input to compute prediction errors (*Jiang and Rao, 2021*; *Keller and Mrsic-Flogel, 2018*). Evidence for this interpretation has mainly come from the discovery of movement-related prediction error responses in a variety of different cortical regions and species (*Attinger et al., 2017*; *Audette et al., 2021*; *Ayaz et al., 2019*; *Eliades and Wang, 2008*; *Heindorf et al., 2018*; *Keller et al., 2012*; *Keller and Hahnloser, 2009*; *Stanley and Miall, 2007*; *Zmarz and Keller, 2016*), and the observation that top-down inputs from motor areas of cortex appear to convey movement-related predictions of sensory input to sensory areas of cortex (*Lein-weber et al., 2017*; *Schneider et al., 2018*). The neurons on which the opposing bottom-up input and top-down prediction converge are referred to as prediction error neurons and are thought to reside primarily in layer 2/3 (L2/3) of cortex (*Jordan and Keller, 2020*). To complete the circuit, there also needs to be a population of neurons that integrate over prediction error responses. These are referred to as internal representation neurons. It is still unclear which cortical neuron type functions as internal representation neurons, but based on anatomical arguments it has been speculated that they can be found in deep cortical layers (*Bastos et al., 2012*).

In humans, externally driven and self-generated activity patterns can be experimentally separated at least partially, for example, by showing subjects a visual stimulus and later asking them to visualize the same stimulus (*Le Bihan et al., 1993*). This type of experiment is considerably more difficult to perform in animals. Using movement as a proxy for self-generated activity patterns, however, using a virtual environment, we can compare a condition in which self-generated and externally generated activity patterns are correlated to a condition in which they are not. In the visual system, we can use a virtual reality environment to experimentally couple or decouple locomotion from visual flow feedback. We will refer to the former as a closed loop condition and to the latter as an open loop condition. Assuming the predictive processing model is correct, we should find differences in the activation patterns in dorsal cortex in response to locomotion onsets as a function of whether they occurred with or without coupled visual flow feedback. Both prediction error neurons and internal representation neurons should respond differentially. Thus, we set out to investigate whether there are differences between closed and open loop activation patterns of mouse dorsal cortex, and whether these differences are neuron type specific. If hallucinations are indeed the result of aberrant activation of internal representations, possibly as a result of misconfigured internal models, antipsychotic drugs should alter the differential activation patterns in these neurons.

## Results

### Tlx3 positive L5 IT neurons distinguished closed and open loop visuomotor coupling

To investigate neuron type-specific differences in activity patterns at locomotion onsets with and without self-generated visual feedback in mouse dorsal cortex, we quantified locomotion onset activity using widefield calcium imaging (*Wekselblatt et al., 2016*). A GCaMP variant was expressed using either a retro-orbital injection of an AAV-PHP.eB vector (*Chan et al., 2017*), or a cross between a Cre line and the Ai148 GCaMP6 reporter line (*Daigle et al., 2018*) (see Methods; *Figure 1A*, *Supplementary file 2*). Mice expressed a GCaMP variant either brain wide (C57BL/6), or in genetically identified subsets of excitatory or inhibitory neurons using Cre driver lines. Please note, we use the term 'brain wide' here to describe an expression pattern that was determined by the intersection of the AAV-PHP.eB tropism and the expression pattern of EF1α or hSyn1 promoters but that was otherwise brain wide (see Discussion). Prior to the start of the imaging experiments, mice were implanted with

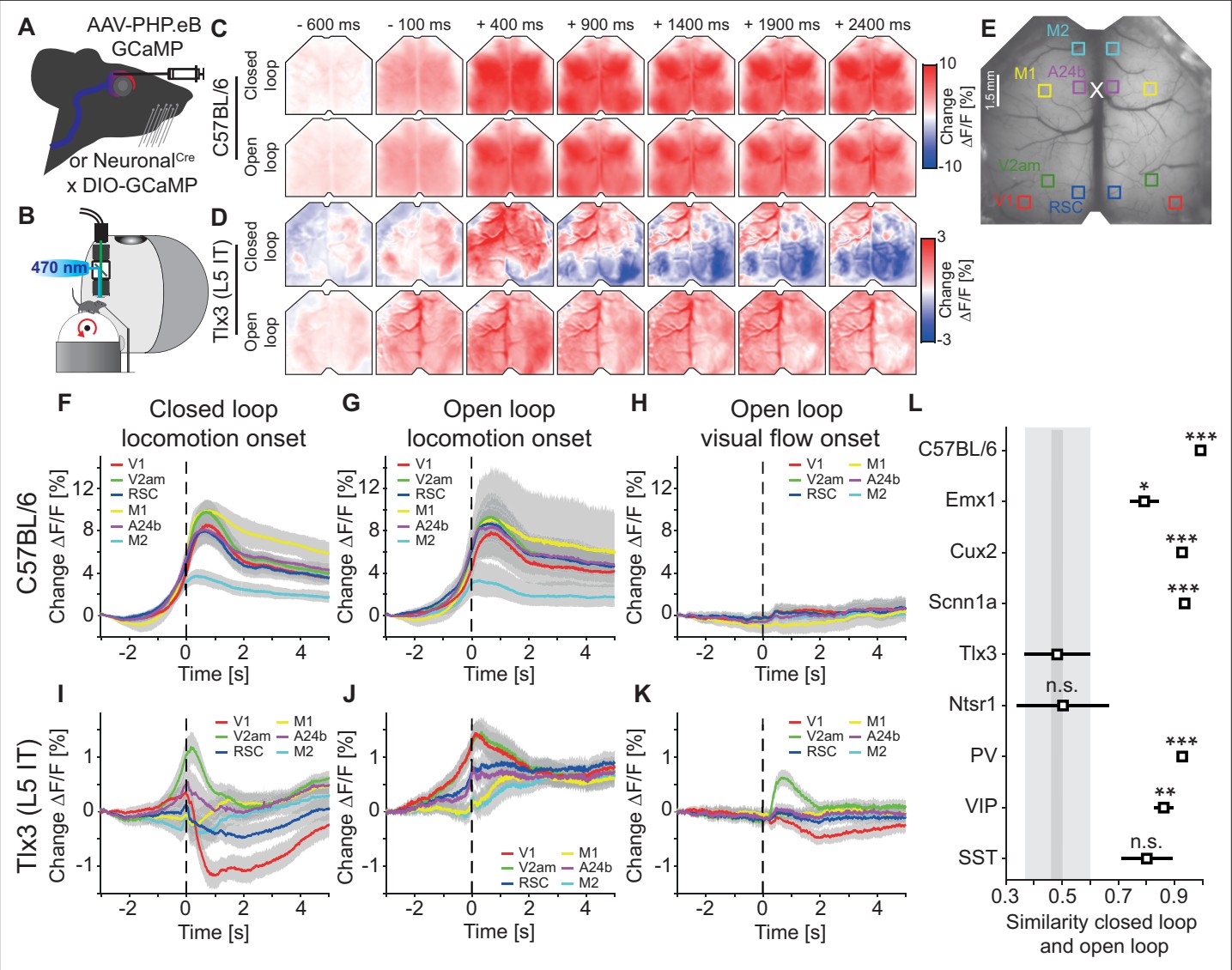

**Figure 1.** Activation patterns in deep cortical layers distinguished closed and open loop locomotion onsets more strongly than superficial layers. (**A**) Schematic of GCaMP expression strategy. We either injected an AAV-PHP.eB virus retro-orbitally to express GCaMP brain wide (C57BL/6), in cortical excitatory neurons (*Emx1*-Cre) or in a subset of SST positive interneurons (see Methods and ***Supplementary file 2***), or used the progeny of a cross of a cell type-specific Cre driver line (Neuronal^Cre^: *Cux2*-CreERT2, *Scnn1a*-Cre, *Tlx3*-Cre, *Ntsr1*-Cre, *PV*-Cre, *VIP*-Cre, or *SST*-Cre) with the Ai148 GCaMP6 reporter line. All mice were then implanted with a crystal skull cranial window prior to imaging experiments. (**B**) Schematic of the experimental setup. We imaged GCaMP fluorescence under 470 nm LED illumination with an sCMOS camera through a macroscope. Mice were free to locomote on an air-supported spherical treadmill while coupled (closed loop), uncoupled (open loop), or no (dark) visual flow in the form of movement along a virtual corridor was displayed on a toroidal screen placed in front of the mouse. Walls of the virtual corridor were patterned with vertical sinusoidal gratings. In a separate condition, we then presented drifting grating stimuli (grating session, see Methods). (**C**) Average response in an example C57BL/6 mouse that expressed GCaMP6 brain wide during closed loop locomotion onsets (top row, 83 onsets) and open loop locomotion onsets (bottom row, 153 onsets). Locomotion onsets in both conditions activated dorsal cortex similarly. (**D**) As in C, but in an example *Tlx3*-Cre × Ai148 mouse that expressed GCaMP6 in layer 5 (L5) intratelencephalic (IT) neurons during closed loop locomotion onsets (top row, 88 onsets) and open loop locomotion onsets (bottom row, 83 onsets). Note that activity decreased in posterior regions of dorsal cortex during closed loop locomotion onsets. (**E**) Example crystal skull craniotomy marking the six regions of interest in each hemisphere we selected: primary visual cortex (V1, red), retrosplenial cortex (RSC, blue), antero-medial secondary visual cortex (V2am, green), primary motor cortex (M1, yellow), anterior cingulate cortex (A24b, purple), and secondary motor cortex (M2, cyan). The white cross marks bregma. (**F**) Average response during closed loop locomotion onsets in C57BL/6 mice that expressed GCaMP brain wide in the six regions of interest (activity was averaged across corresponding regions in both hemispheres). Mean (lines) and 90% confidence interval (shading) are calculated as a hierarchical bootstrap estimate for each time bin (see Methods and ***Supplementary file 1***). (**G**) As in F, but for open loop locomotion onsets. (**H**) As in F, but for visual flow onsets in the open loop condition restricted to times when the mice were not locomoting. (**I**) Average response during closed loop locomotion onsets in *Tlx3*-Cre × Ai148 mice that expressed GCaMP6 in L5 IT neurons, activity was averaged

*Figure 1 continued on next page*

*Figure 1 continued*

across corresponding regions in both hemispheres. Mean (lines) and 90% confidence interval (shading) are calculated as a hierarchical bootstrap estimate for each time bin (see Methods and *Supplementary file 1*). (J) As in I, but for open loop locomotion onsets. (K) As in J, but for visual flow onsets during open loop sessions restricted to times when the mice were not locomoting. (L) Similarity of the average closed and open loop locomotion onset responses quantified as the correlation coefficient between the two in a window −5 s to +3 s around locomotion onset (see Methods). Error bars indicate SEM over the 12 (6 per hemisphere) cortical regions. Statistical comparisons are against the *Tlx3* data and were corrected for family-wise error rate (see Methods and *Supplementary file 1*): adjusted significance thresholds, n.s.: not significant, *: p<0.05/9, **: p<0.01/9, ***: p<0.001/9.

The online version of this article includes the following figure supplement(s) for figure 1:

**Figure supplement 1.** Fluorescence changes driven by hemodynamic occlusion.

**Figure supplement 2.** In primary visual cortex (V1), the sum of locomotion and visual flow onset could not explain the closed loop locomotion onset response of layer 5 (L5) intratelencephalic (IT) neurons.

**Figure supplement 3.** Layer 5 (L5) intratelencephalic (IT) soma recorded in primary visual cortex (V1) with two-photon imaging show similar patterns of activity as the widefield signal recorded at the surface of the dorsal cortex.

**Figure supplement 4.** Layer 5 (L5) intratelencephalic (IT) neurons had strikingly different responses during closed and open loop locomotion onsets compared to other cortical neuron types.

a crystal skull cranial window (*Kim et al., 2016*). We used crystal skull preparations as we found that hemodynamic artifacts were smaller in this preparation compared to those observed in clear skull preparations (*Figure 1—figure supplement 1A and B*). For all experiments, mice were head-fixed and free to locomote on a spherical treadmill (*Figure 1B*). Activity from dorsal cortex was imaged while mice were in a closed loop visuomotor condition, in which locomotion on the spherical treadmill controlled forward movement in a virtual corridor. Following this, mice were exposed to an open loop condition during which mice were presented with a replay of the visual flow they had self-generated in the preceding closed loop condition.

To test whether there are differences between closed and open loop locomotion onset responses, we first compared these in mice that expressed GCaMP brain wide (C57BL/6). Consistent with previous findings (*Musall et al., 2019*; *Stringer et al., 2019*), we found that locomotion onset resulted in broad activation across most of dorsal cortex in both conditions (*Figure 1C*). However, comparing locomotion onsets in mice that expressed GCaMP6 only in *Tlx3* positive layer 5 (L5) IT neurons, we found that the activation pattern was strikingly different between closed and open loop conditions (*Figure 1D*). While locomotion onset in the open loop condition resulted in an increase in activity across dorsal cortex, similar to that observed in C57BL/6 mice with brain wide GCaMP expression, locomotion onset in the closed loop condition resulted in a very different activation pattern: In posterior parts of dorsal cortex, the activity decreased, while in anterior parts of the dorsal cortex, the activity increased. To quantify these activation patterns across mice, we selected six regions of interest (ROIs): primary visual cortex (V1), antero-medial secondary visual cortex (V2am), retrosplenial cortex (RSC), primary motor cortex (M1), anterior cingulate cortex at the level of bregma (A24b), and secondary motor cortex (M2) (*Figure 1E*). In C57BL/6 mice that expressed GCaMP brain wide, closed loop locomotion onset resulted in similar activity patterns across these six regions (*Figure 1F*). This activity pattern was not markedly different from that observed during locomotion onsets in an open loop condition (*Figure 1G*). To quantify the contribution of visual flow onset to the closed loop locomotion onset activity, we measured responses to visual flow onsets that occurred in the absence of locomotion in the open loop condition (*Figure 1H*). In C57BL/6 mice with brain wide GCaMP expression, visual flow onsets resulted in responses that were small compared to those observed during locomotion onset, even in V1. Note that a visual flow onset is a relatively weak visual stimulus compared to a drifting grating stimulus. A grating stimulus is typically composed of a gray screen or a standing grating stimulus that suddenly transitions to a drifting grating, and thus the visual flow undergoes a very high acceleration. A visual flow onset in an open loop condition is the replay of previously self-generated visual flow and thus exhibits much lower visual flow acceleration. The absence of a strong visual flow onset response in C57BL/6 mice that expressed GCaMP brain wide is consistent with a high similarity of closed and open loop locomotion onset responses.

Different from what we found in mice that expressed GCaMP brain wide, in mice that expressed GCaMP6 in *Tlx3* positive L5 IT neurons, the closed loop locomotion onset patterns differed markedly when compared across the six ROIs (*Figure 1I*). Moreover, these activation patterns were different from those observed during open loop locomotion onsets (*Figure 1J*). While these responses could

partially be explained by hemodynamic occlusion, the time course of hemodynamic occlusion in *Tlx3*-Cre mice that expressed eGFP in L5 IT neurons was similar to that observed in C57BL/6 mice that expressed eGFP brain wide and mainly resulted in a decrease of fluorescence (***Figure 1—figure supplement 1C and D***). In *Tlx3* positive L5 IT neurons, open loop visual flow onsets that occurred in the absence of locomotion resulted in quantifiable responses in V1 and V2am (***Figure 1K***), but the closed loop locomotion onset responses in V1 could not be explained by the sum of locomotion and visual flow onset responses (***Figure 1—figure supplement 2A and C***). Most of the widefield signals we record here likely originate in compartments in superficial layers which are composed of axons and dendrites of local neurons, and long-range inputs (***Allen et al., 2017***) (see Discussion). To test how well the widefield signals recorded in V1 reflected local somatic activity in V1, we repeated the experiments using two-photon calcium imaging of somatic activity in *Tlx3* positive L5 IT neurons. We found that while there were differences between the responses measured with widefield imaging and those measured at the soma, both for locomotion onset responses and for visual responses, these differences did not change our conclusions (***Figure 1—figure supplement 3***).

To quantify the differences between closed and open loop locomotion onset responses of L5 IT neurons in relation to those observed in other cortical neuron types, we repeated the experiments for the selection of excitatory and inhibitory cortical neuron types (***Figure 1—figure supplement 4A–I***). For each neuron type, we quantified the similarity of the activity patterns associated with closed and open loop locomotion onsets by calculating the correlation coefficient between the activity traces in a window around locomotion onset (–5 s to +3 s) for each ROI (see Methods). Quantifying the differences in V1, we find that the similarity in *Tlx3* positive L5 IT neurons was lower than that observed in all other neuron types tested, except for *Ntsr1* positive L6 neurons (***Figure 1—figure supplement 4J***). *Tlx3* positive L5 IT neurons and *Ntsr1* positive L6 neurons also exhibit a lower similarity than other neuron types when comparing the similarity across all ROIs (***Figure 1L***). Both in V1 across mice, as well as when comparing across all areas, we found the highest similarity between closed and open loop locomotion onsets across regions in C57BL/6 mice that expressed GCaMP brain wide. Thus, among the neuron types tested, excitatory neurons of deep cortical layers exhibited the largest differences between closed and open loop locomotion-related activation across all the selected cortical regions. The remarkable aspect of these findings is not that there are differences between closed and open loop locomotion onsets, particularly in visual regions of cortex one would perhaps expect to find differences, but that these differences are larger specifically in the deep cortical layers.

To quantify how the interaction between locomotion and visual flow feedback changes across dorsal cortex, we computed correlations between calcium activity and locomotion speed for each ROI, for closed and open loop conditions separately. In C57BL/6 mice that expressed GCaMP brain wide, this correlation was comparable between closed and open loop conditions, was highest in posterior regions of dorsal cortex, and lower in more anterior regions (***Figure 2A and B***). In mice that expressed GCaMP6 in *Tlx3* positive L5 IT neurons, the correlation between locomotion speed and calcium activity in the closed loop condition was negative in posterior regions of dorsal cortex and positive in anterior regions. Conversely, in open loop conditions, the correlation was positive throughout dorsal cortex (***Figure 2C and D***). To test whether it was the coupling of visual flow that resulted in a reduction of the correlation in the closed loop condition or whether the presence of an uncoupled visual stimulus resulted in an increase of correlation in the open loop condition, we also compared these correlations to those observed when mice were locomoting in relative darkness (see Methods). In C57BL/6 mice that expressed GCaMP brain wide, the correlation of calcium activity with locomotion in darkness was indistinguishable from that in closed or open loop (***Figure 2B***). In mice that expressed GCaMP6 in *Tlx3* positive L5 IT neurons, an interesting pattern emerged. In visual regions of dorsal cortex, the correlation between locomotion speed and calcium activity in relative darkness resembled that observed in open loop conditions, while in frontal regions, it resembled that observed in closed loop conditions. In visual regions, the coupling of visual flow resulted in a reduction of correlation, while in frontal regions the presence of an uncoupled visual stimulus resulted in an increase of correlations. Thus, the difference between closed and open loop conditions likely had different origins in visual and frontal regions of dorsal cortex. Overall, we concluded that the coupling of locomotion and resulting visual flow feedback not only activated *Tlx3* positive L5 IT neurons differentially but did so throughout dorsal cortex, well beyond visual regions.

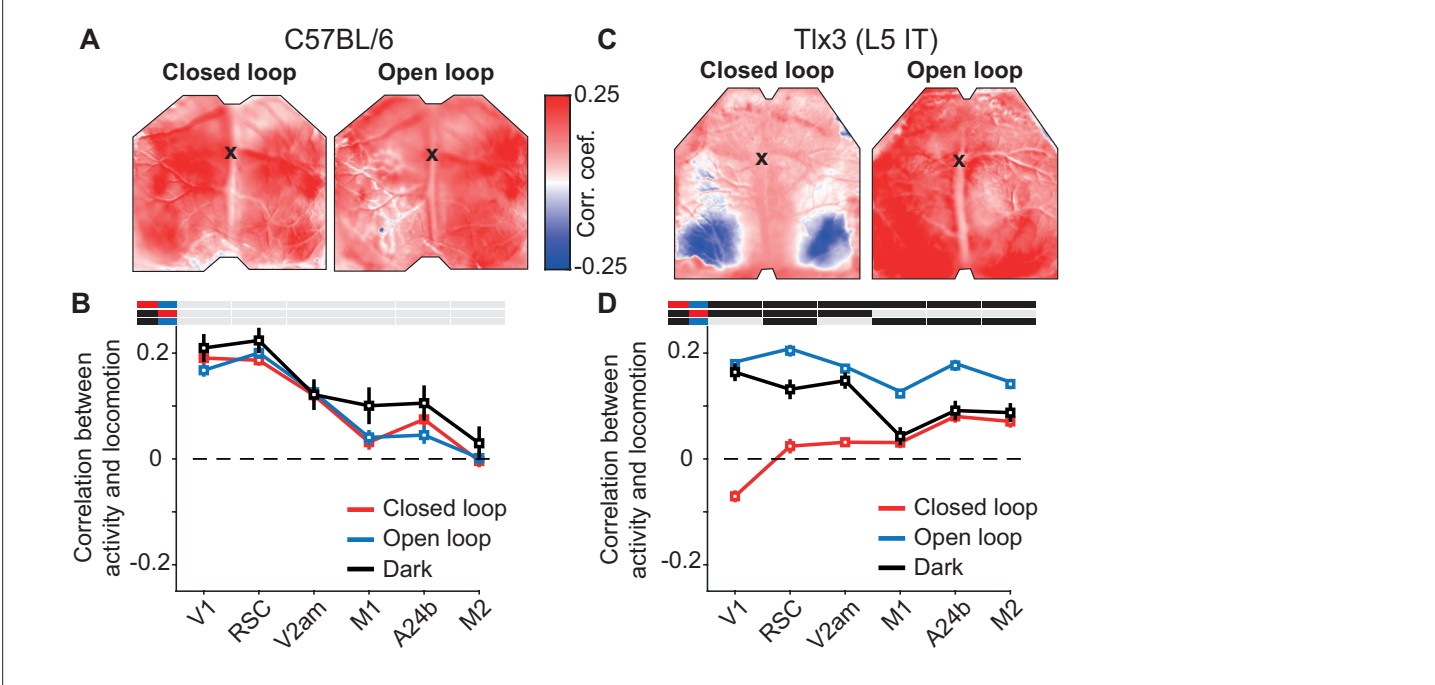

**Figure 2.** Layer 5 (L5) intratelencephalic (IT) neurons were differentially activated by locomotion depending on the type of visual feedback.
(**A**) Correlation between calcium activity and locomotion speed in the closed loop (left) and open loop (right) conditions, calculated for each pixel in the image for one example C57BL/6 mouse that expressed GCaMP brain wide. The black cross marks bregma. Activity in most of dorsal cortex correlates positively with locomotion speed in both closed and open loop conditions. (**B**) Average correlation between calcium activity and locomotion speed in the six regions of interest in closed loop (red), open loop (blue), or dark (black) conditions in six C57BL/6 mice that expressed GCaMP brain wide (308, 316, and 68 five-min recording sessions, respectively). Error bars indicate SEM over recording sessions. Bars above the plot indicate significant differences between conditions (compared conditions are indicated by colored line segments to the left, black: p<0.05, gray: n.s., see **Supplementary file 1** for all information on statistical testing). On average, the correlation was highest in posterior dorsal cortex and was not different between conditions. (**C**) As in A, but for an example *Tlx3*-Cre × Ai148 mouse that expressed GCaMP6 in L5 IT neurons. Correlations of calcium activity and locomotion speed were lower in the closed loop condition compared to the open loop condition, most prominently in posterior regions of dorsal cortex. (**D**) As in B, but for 15 *Tlx3*-Cre × Ai148 mice that expressed GCaMP6 in L5 IT neurons (data from 420 closed loop, 394 open loop, and 194 dark 5 min recording sessions). Error bars indicate SEM over recording sessions. Correlation differed significantly between the closed and the open loop condition and was lowest in posterior dorsal cortex during the closed loop condition.

## Visuomotor prediction errors originated in visual cortical regions and had opposing effects on *Tlx3* positive L5 IT activity

A consequence of the natural coupling of locomotion and visual flow feedback is that movements are a good predictor of self-generated sensory feedback. In our closed loop experiments, forward locomotion was a strong predictor of backward visual flow. It has been speculated that such predictions are used to compute prediction errors in L2/3 of cortex (*Jordan and Keller, 2020*). Prediction errors come in two variants: Positive prediction errors that signal more visual flow than predicted and negative prediction errors that signal less visual flow than predicted (*Keller and Mrsic-Flogel, 2018*; *Rao and Ballard, 1999*). To probe for negative and positive prediction error signals, we quantified the responses to visuomotor mismatch events and to unexpected visual stimuli (representing less and more visual flow input than expected, respectively). Visuomotor mismatch was introduced by briefly (1 s) halting visual flow during self-generated feedback in the closed loop condition. Visual responses were elicited by presenting full-field (covering the entire toroidal screen) drifting grating stimuli to a passively observing mouse (see Methods). Please note, both types of prediction errors can be probed using a variety of stimuli, but the two we are using here, in our experience, are the most robust at eliciting strong responses in V1. In C57BL/6 mice that expressed GCaMP brain wide, both visuomotor mismatch and grating stimuli resulted in increases of activity that were strongest and appeared first in visual regions of dorsal cortex (*Figure 3A–C* and *Figure 1—figure supplement 2B*). In mice that expressed GCaMP6 in *Tlx3* positive L5 IT neurons, visuomotor mismatch resulted in an

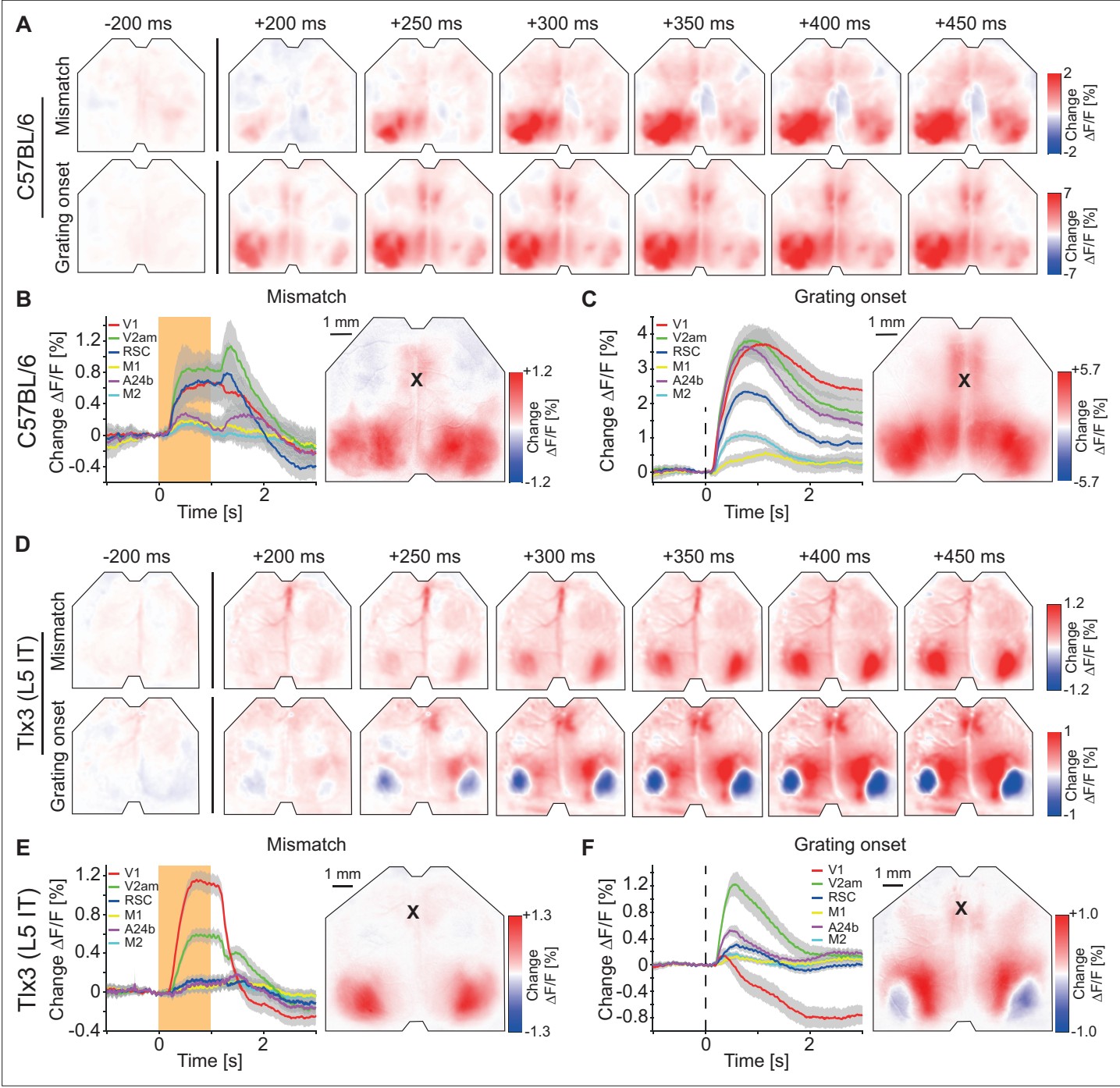

**Figure 3.** Visuomotor prediction error responses in dorsal cortex originated in primary visual cortex (V1) and activated layer 5 (L5) intratelencephalic (IT) neurons differentially. (**A**) Average responses to mismatches (top row, 230 onsets) and drifting gratings (bottom row, 86 onsets) in an example C57BL/6 mouse that expressed GCaMP brain wide. (**B**) Left: Average response to mismatches in C57BL/6 mice that expressed GCaMP brain wide in six cortical regions (activity was averaged across corresponding regions in both hemispheres). Mean (lines) and 90% confidence interval (shading) are calculated as a hierarchical bootstrap estimate for each time bin (see Methods and *Supplementary file 1*). Orange shading indicates duration of mismatch event. Right: Average response map to mismatch in six C57BL/6 mice (see Methods). Black cross marks bregma. (**C**) As in B, but for drifting grating responses. (**D**) As in A, but in an example *Tlx3*-Cre × Ai148 mouse that expressed GCaMP6 in L5 IT neurons (top row: mismatch, 292 onsets; bottom row: drifting gratings, 171 onsets). (**E**) As in B, but for *Tlx3*-Cre × Ai148 mice that expressed GCaMP6 in L5 IT neurons. (**F**) As in C, but for *Tlx3*-Cre × Ai148 mice that expressed GCaMP6 in L5 IT neurons.

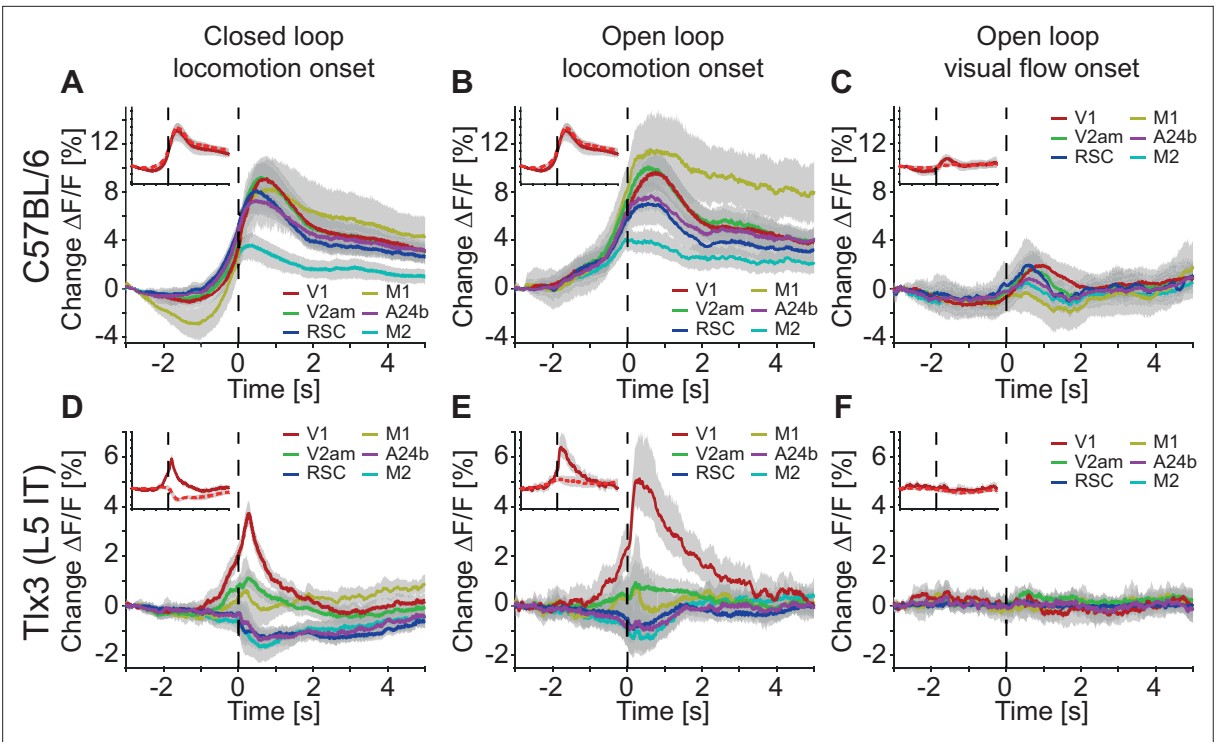

**Figure 4.** Clozapine increased locomotion-related responses in layer 5 (L5) intratelencephalic (IT) neurons in primary visual cortex (V1). (**A**) Average response during closed loop locomotion onsets in C57BL/6 mice that expressed GCaMP brain wide (activity was averaged across corresponding regions in both hemispheres) after a single injection of the antipsychotic drug clozapine (see Methods). Mean (lines) and 90% confidence interval (shading) are calculated as a hierarchical bootstrap estimate for each time bin (see Methods and **Supplementary file 1**). Inset: Comparison of responses for V1 in naive C57BL/6 mice (dashed red) and the same mice after clozapine injection (dark red, same data as in main panel). (**B**) As in A, but for open loop locomotion onsets after a clozapine injection. (**C**) As in A, but for open loop visual flow onsets restricted to times when the mice were not locomoting. (**D**) Average response during closed loop locomotion onsets in *Tlx3*-Cre × Ai148 mice that expressed GCaMP6 in layer L5 IT neurons (activity was averaged across corresponding regions in both hemispheres) after a single injection of the antipsychotic drug clozapine (see Methods). Mean (lines) and 90% confidence interval (shading) are calculated as a hierarchical bootstrap estimate for each time bin (see Methods and **Supplementary file 1**). Inset: Comparison of responses for V1 in naive *Tlx3-Cre × Ai148* mice (dashed red) and the same mice after clozapine injection (dark red, same data as in main panel). (**E**) As in D, but for open loop locomotion onsets. (**F**) As in D, but for open loop visual flow onsets restricted to times when the mice were not locomoting.

The online version of this article includes the following figure supplement(s) for figure 4:

**Figure supplement 1.** Additional information for antipsychotic drug data.

increase of activity in V1, while grating onsets resulted in a decrease of activity in V1 (**Figure 3D–F** and **Figure 1—figure supplement 2D**). Thus, in dorsal cortex, both mismatch and drifting grating stimuli primarily activated visual cortex but resulted in opposing activation patterns in *Tlx3* positive L5 IT neurons. An increase in activity to negative prediction errors and a decrease to positive prediction errors would be consistent with either a population of negative prediction error neurons or a type of internal representation neuron (**Rao and Ballard, 1999**). Given that L5 neurons in V1 do not exhibit the opposing influence of top-down and bottom-up inputs necessary to compute prediction errors (**Jordan and Keller, 2020**), we speculated that *Tlx3* positive L5 IT neurons might function as a type of internal representation neuron.

## The antipsychotic drug clozapine alters visuomotor integration in L5

The activation of an internal representation is the brain's best guess at what the stimuli in the environment are. We speculated that the activation of such an internal representation would be particularly susceptible to substances that are known to reduce illusory percepts, like antipsychotic drugs. We thus quantified the changes in dorsal cortex activity associated with a single intraperitoneal injection of an antipsychotic drug (clozapine). In C57BL/6 mice that expressed GCaMP brain wide, we found that closed and open loop locomotion onsets were almost unaffected by clozapine (**Figure 4A and**

*B*), while there was a small increase in open loop visual flow onset responses (*Figure 4C*). In mice that expressed GCaMP6 in *Tlx3* positive L5 IT neurons, however, clozapine fundamentally changed both closed and open loop locomotion onset responses. In V1, both types of locomotion onset now resulted in a massive increase in activity (*Figure 4D and E*). Conversely, open loop visual flow onset responses remained largely unchanged (*Figure 4F*). These effects could not be explained by clozapine-induced changes in hemodynamic occlusion (*Figure 4—figure supplement 1A*) or locomotion behavior (*Figure 4—figure supplement 1B and C*). Clozapine also had an opposing effect on the average correlation of activity with locomotion speed in *Tlx3* positive L5 IT neurons. While the average correlation between activity and locomotion speed was increased in C57BL/6 mice that expressed GCaMP brain wide, the same measure was decreased in mice that expressed GCaMP6 in *Tlx3* positive L5 IT neurons (*Figure 4—figure supplement 1D*). Finally, we confirmed that this effect was also present when recording somatic activity in V1. Using two-photon imaging, after clozapine administration we observed an increase of somatic responses during locomotion onsets that was very similar to that observed with widefield imaging (*Figure 4—figure supplement 1E and F*). Thus, consistent with the speculation that *Tlx3* positive L5 IT neurons might function as internal representation neurons, clozapine exhibited a substantially stronger influence on locomotion onset responses in *Tlx3* positive L5 IT neurons than would be expected from the effect of clozapine on brain wide responses.

## Antipsychotic drugs decouple long-range cortico-cortical activity

L5 IT neurons are likely the primary source of long-range cortical communication (*Chen et al., 2019*; *Harris et al., 2019*; *Leinweber et al., 2017*). Given that clozapine increased locomotion onset responses in L5 IT neurons in V1, we wondered whether clozapine might more generally change the long-range influence between L5 IT neurons. To investigate this possibility, we computed the correlations of calcium activity between the six ROIs across both hemispheres and all recording conditions. We did this first in C57BL/6 mice that expressed GCaMP brain wide before (*Figure 5A*) and after a single intraperitoneal injection of clozapine (*Figure 5B*). Overall, clozapine resulted in a relatively modest decrease in correlations. On repeating this experiment in mice that expressed GCaMP6 in *Tlx3* positive L5 IT neurons, we found a larger decrease in correlations that appeared to be stronger for regions that were further apart (*Figure 5C and D*). To quantify these changes, we plotted the correlations as a function of linear distance between the regions as measured in a top-down view of dorsal cortex (*Figure 5—figure supplement 1A*). Note, this is only an approximation of the actual axonal path length between the regions. To visualize the distribution of correlations, we interpolated the data and represented them as heatmaps (*Figure 5—figure supplement 1B*). The resulting representation for the data from C57BL/6 mice that expressed GCaMP brain wide before and after clozapine injection is shown in *Figure 6A and B*, respectively. We then split the data into approximately equal portions of short- and long-range correlations using a cutoff of 0.9 times the bregma-lambda distance (approximately 3.8 mm). While both short- and long-range correlations were reduced, this reduction was not significant in either case (*Figure 6C*). We also found no evidence of a difference between short- and long-range correlation changes. In mice that expressed GCaMP6 in *Tlx3* positive L5 IT neurons, however, we found a significant reduction in both short- and long-range correlations (*Figure 6D and E*), and both were more strongly reduced than in C57BL/6 mice that expressed GCaMP brain wide (C57BL/6 vs *Tlx3*-Cre × Ai148, short-range: $p < 0.05$; long-range: $p < 10^{-5}$; rank-sum test). Moreover, the reduction in activity correlations was stronger for long-range correlations than it was for short-range correlations (*Figure 6F*). To control for possible clozapine-induced changes in hemodynamic occlusion, we repeated the experiment and analyses in *Tlx3*-Cre mice that expressed eGFP in L5 IT neurons and found that alterations in hemodynamic occlusion could not account for the decrease in correlations but resulted in a small increase of the same metric (*Figure 5—figure supplement 1C–E*). To test whether the stress of injection, or simply passage of time, could explain the observed effects, we conducted a separate set of experiments in which we injected saline in a cohort of *Tlx3*-Cre × Ai148 mice and performed the same analysis. We found no evidence of a change in correlations with saline injections (*Figure 5—figure supplement 1F–H*). To further dissect the spatial scale at which this decorrelation occurred, we repeated the experiments in *Tlx3*-Cre × Ai148 mice that expressed GCaMP6 in L5 IT neurons, recording calcium activity using both a widefield and a two-photon microscope. We found that while the activity correlations measured with the widefield imaging decreased (*Figure 6—figure supplement 1A–C*), pairwise somatic activity correlations measured by two-photon

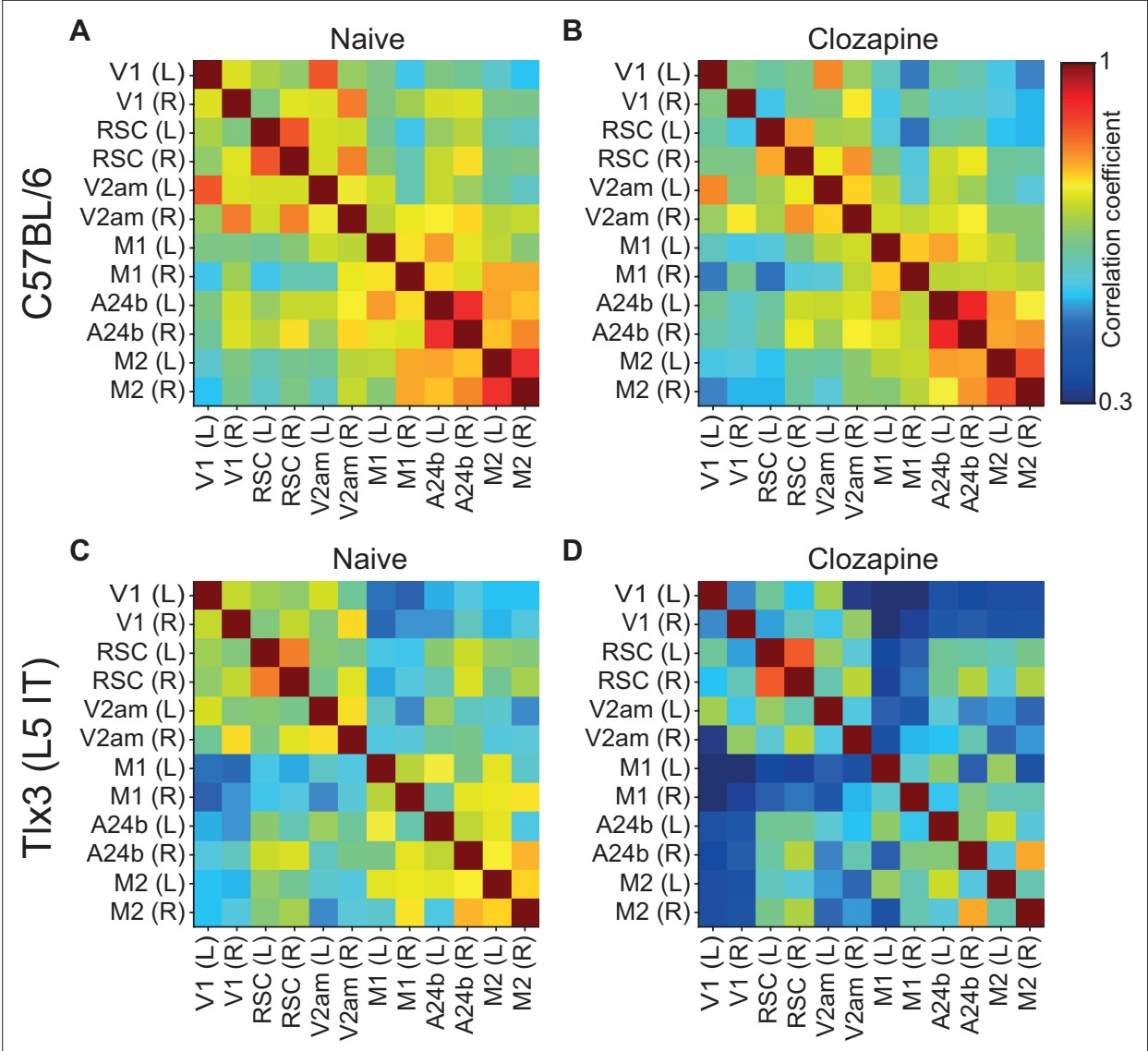

**Figure 5.** Clozapine reduced activity correlations in dorsal cortex predominantly in layer 5 (L5) intratelencephalic (IT) neurons. (**A**) Average correlation of activity between the 12 regions of interest in four C57BL/6 mice that expressed GCaMP brain wide. (**B**) As in A, but after a single injection of the antipsychotic drug clozapine in the same four C57BL/6 mice. (**C**) Average correlation of activity between the 12 regions of interest in five *Tlx3*-Cre × Ai148 mice that expressed GCaMP6 in layer L5 IT neurons. (**D**) As in C, but after a single injection of the antipsychotic drug clozapine in the same five *Tlx3*-Cre × Ai148 mice.

The online version of this article includes the following figure supplement(s) for figure 5:

**Figure supplement 1.** Calculation of the distance-correlation heatmaps.

imaging did not (*Figure 6—figure supplement 1D*). This is consistent with the idea that decorrelation primarily occurs for long-range interactions. Finally, to confirm that the clozapine-induced reduction in correlation was stronger in deep cortical layers, we repeated the experiment in a population of L2/3 and L4 excitatory neurons using *Cux2*-CreERT2 × Ai148 mice. We found that clozapine also decreased correlations of cortical activity in superficial excitatory neurons (*Figure 6—figure supplement 2*). However, this reduction was significantly weaker than the reduction we observed in mice that expressed GCaMP6 in *Tlx3* positive L5 IT neurons (*Cux2*-CreERT2 × Ai148 vs *Tlx3*-Cre × Ai148, short-range: p<0.005, long-range: p<10⁻⁸, rank-sum test). Thus, administration of the antipsychotic drug clozapine resulted in a decorrelation of activity across dorsal cortex that was stronger in L5 IT neurons (*Tlx3*) than it was in either upper layer excitatory neurons (*Cux2*) or in the brain wide average (C57BL/6).

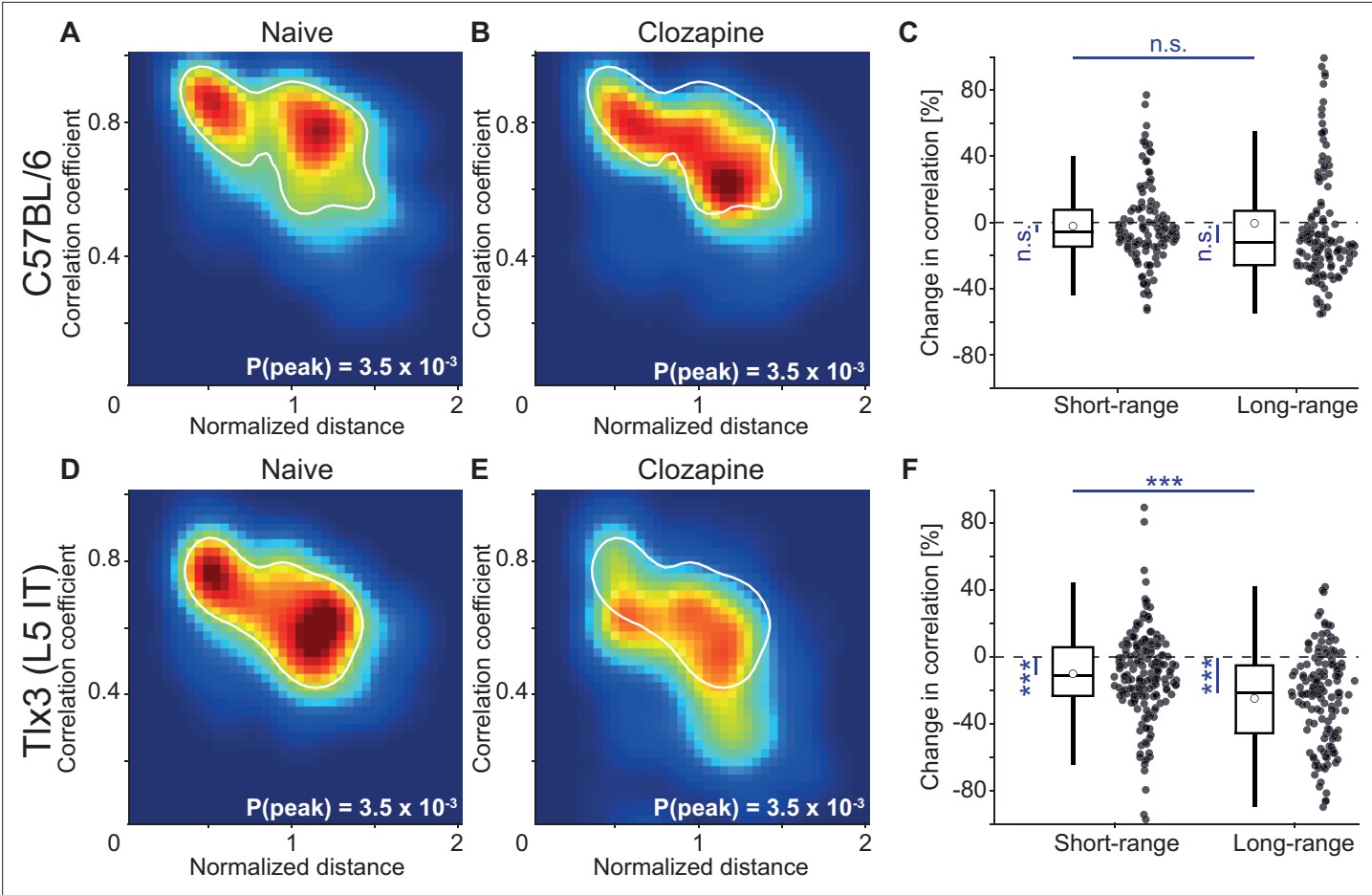

**Figure 6.** Layer 5 (L5) intratelencephalic (IT) neurons exhibited the strongest clozapine-induced reduction of long-range correlations. (**A**) Density map of correlation coefficients as a function of distance between the regions, normalized across mice by the bregma-lambda distance (see **Figure 5—figure supplement 1A and B**, and Methods), for four naive C57BL/6 mice that expressed GCaMP brain wide. Peak density is indicated at the bottom right of the plot and corresponds to the maximum value of the color scale (dark red). The white line is a contour line drawn at 50% of peak value. (**B**) As in A, but for the same four C57BL/6 mice after a single injection of the antipsychotic drug clozapine. The white line is the contour line from panel A for comparison. (**C**) Clozapine-induced change in the activity correlation between regions, normalized to the correlation coefficient in the naive state, in four C57BL/6 mice. Data were split into short- and long-range activity correlations (see Methods). Boxes represent the median and the interquartile range. The open circle indicates the mean of the distribution. The whiskers mark 1.5 times the interquartile range. Dots are the individual data points (short-range: 122 pairs of regions, long-range: 142 pairs of regions). n.s.: not significant. See **Supplementary file 1** for all information on statistical testing. (**D**) As in A, but for five *Tlx3*-Cre × Ai148 mice that expressed GCaMP6 in L5 IT neurons. (**E**) As in B, but for five *Tlx3*-Cre × Ai148 mice that expressed GCaMP6 in L5 IT neurons. The white line is the contour line from panel D for comparison. (**F**) As in C, but for 5 *Tlx3*-Cre × Ai148 mice that expressed GCaMP6 in L5 IT neurons (short-range: 182 pairs of regions, long-range: 148 pairs of regions). ***: p<0.001.

The online version of this article includes the following figure supplement(s) for figure 6:

**Figure supplement 1.** The clozapine-induced decorrelation was not present in local somatic correlations.

**Figure supplement 2.** The clozapine-induced decorrelation of dorsal cortex activity was weaker in superficial cortical layers than in layer 5 (L5).

To test whether the decorrelation of the activity of L5 IT neurons is specific to clozapine or might more generally be a functional signature of antipsychotic drugs, we repeated the experiments with two additional antipsychotic drugs, aripiprazole and haloperidol. We found that the principal effect of decorrelation of the activity patterns of *Tlx3* positive L5 IT neurons was preserved with both drugs (**Figure 7A–F**). These changes in correlation were absent upon injection of a psychostimulant, amphetamine (**Figure 7G–I**). Thus, it is possible that a decrease in long-range correlations between cortical L5 IT neurons might be a common mechanism of action for antipsychotic drugs.

Changes in the correlation between cortical regions could either be driven by changes in correlated external inputs or through changes in communication between cortical regions. If antipsychotic drugs

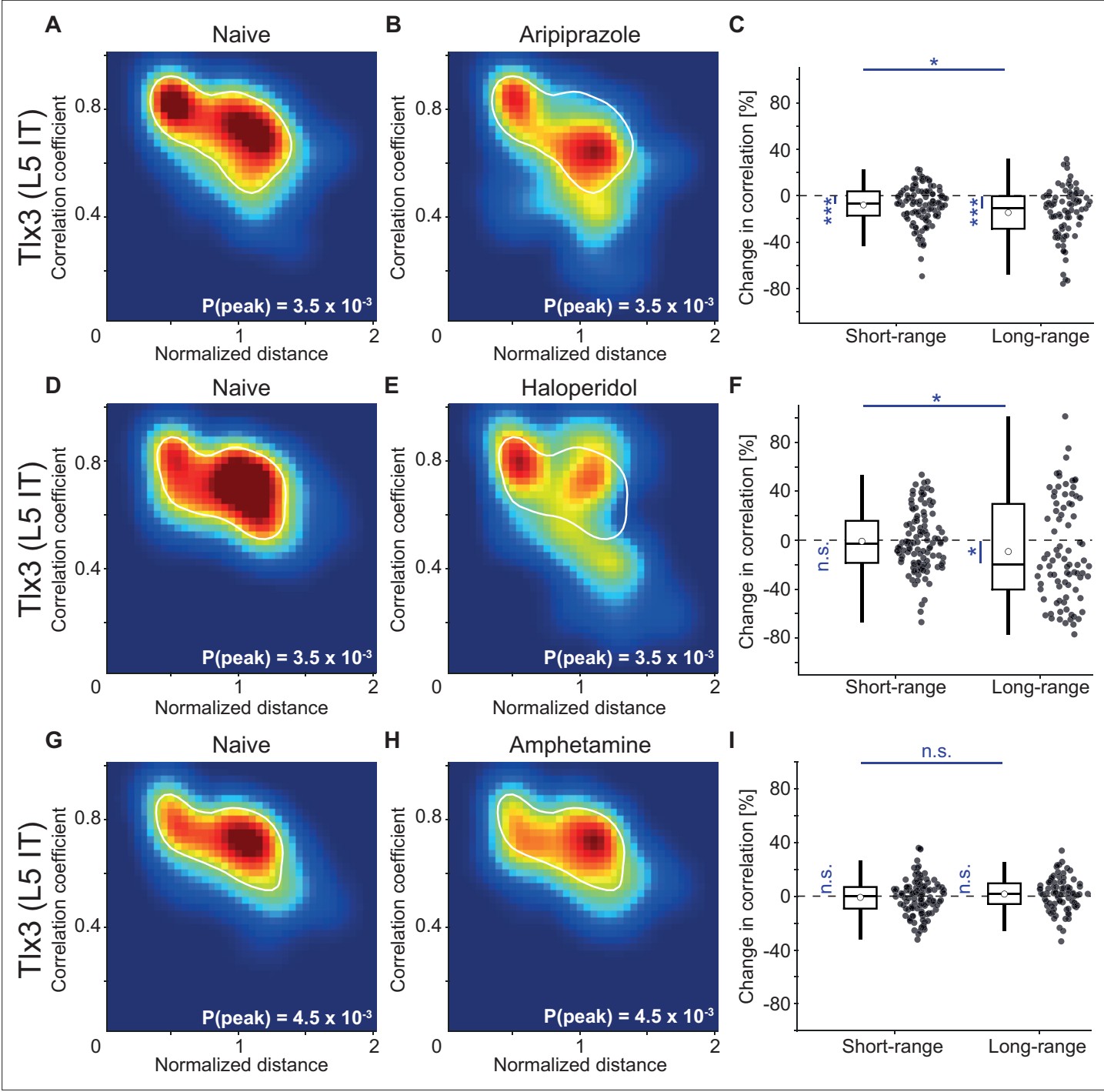

**Figure 7.** Antipsychotic drugs aripiprazole and haloperidol mimicked the decorrelation of layer 5 (L5) intratelencephalic (IT) activity observed with clozapine while psychostimulant amphetamine did not. (**A**) Density map of correlation coefficients as a function of distance between the regions, normalized across mice by the bregma-lambda distance (see *Figure 5—figure supplement 1A and B*, and Methods), for three *Tlx3*-Cre × Ai148 mice that expressed GCaMP6 in L5 IT neurons. Peak density is indicated at the bottom right of the plot and corresponds to the maximum value of the color scale (dark red). The white line is a contour line drawn at 50% of peak value. (**B**) As in A, for the same three *Tlx3*-Cre × Ai148 mice, but after a single injection of the antipsychotic drug aripiprazole. The white line is the contour line from panel A for comparison. (**C**) Aripiprazole-induced change in the activity correlation between regions, normalized to the correlation coefficient in the naive state, in three *Tlx3*-Cre × Ai148 mice. Data were split into short- and long-range activity correlations (see Methods). Boxes represent the median and the interquartile range. The open circle indicates the mean of the distribution. The whiskers mark 1.5 times the interquartile range. Dots are the individual data points (short-range: 114 pairs of regions, long-range: 84 pairs of regions). *: p<0.05, ***: p<0.001. See *Supplementary file 1* for all information on statistical testing. (**D**) As in A, but for three different

*Figure 7 continued on next page*

*Figure 7 continued*

*Tlx3*-Cre × Ai148 mice. (**E**) As in D, for the same three *Tlx3*-Cre × Ai148 mice, but after a single injection of the antipsychotic drug haloperidol. The white line is the contour line from panel D for comparison. (**F**) As in C, but for the three mice that had received a single injection of the antipsychotic drug haloperidol (short-range: 114 pairs of regions, long-range: 84 pairs of regions). *: p<0.05; n.s.: not significant. (**G**) As in A, but for three different *Tlx3*-Cre × Ai148 mice. (**H**) As in G, for the same three *Tlx3*-Cre × Ai148 mice, but after a single injection of the psychostimulant amphetamine. The white line is the contour line from panel G for comparison. (**I**) As in D, but for the three mice that had received a single injection of the psychostimulant amphetamine (short-range: 114 pairs of regions, long-range: 84 pairs of regions). n.s.: not significant.

primarily reduce correlations by reducing the strength of communication between cortical regions, independent of whether this communication is direct or indirect, we would expect to find changes in the way responses spread between cortical regions. We used visuomotor prediction error responses that originated in V1 (*Figure 3*) to quantify how the spread of these responses in *Tlx3* positive L5 IT neurons was influenced by antipsychotic drugs. The computation of negative prediction errors during movement is thought to be driven by an excitatory prediction of visual flow mediated by long-range cortical input from L5 neurons in regions like A24b (*Leinweber et al., 2017*), as well as an inhibitory bottom-up visual input. By contrast, the computation of a positive prediction error during passive observation is thought to be driven by an excitatory bottom-up visual input and an inhibitory top-down input. Thus, we would primarily expect to find a reduction in mismatch responses in L5 IT neurons and a reduction in the spread of this signal to secondary visual regions. This is indeed what we observed: Mismatch responses were partially reduced in V1 and almost absent in V2am after injection of antipsychotic drugs (*Figure 8A*). While responses to onsets of drifting gratings were also reduced in V2am, they were overall less affected by the antipsychotic drugs than mismatch responses (*Figure 8B*). These results are consistent with the interpretation that antipsychotic drugs reduce lateral communication in L5. Whether this communication is mediated by direct long-range connections between L5 neurons, or indirectly via other cortical neurons or subcortical loops, remains to be tested.

## Discussion

When interpreting our results, it should be kept in mind that the source of the widefield imaging signal in these experiments is not entirely clear and is likely a combination of somatic, dendritic, and axonal signals. Given that the signal attenuation half depth in cortex is on the order of 100 μm (*Aravanis*

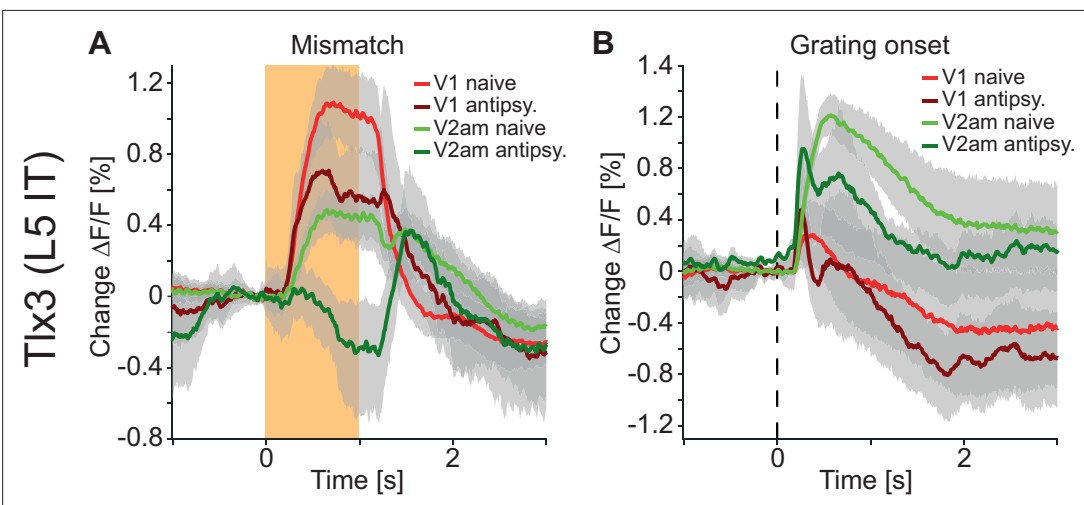

**Figure 8.** Antipsychotic drug treatment preferentially reduced responses to and propagation of negative prediction errors. (**A**) Average responses to mismatches in *Tlx3*-Cre × Ai148 mice that expressed GCaMP6 in layer 5 (L5) intratelencephalic (IT) neurons before (red: primary visual cortex [V1] naive, green: antero-medial secondary visual cortex [V2am] naive, activity was averaged across corresponding regions in both hemispheres) and after (dark red: V1, dark green: V2am) injection of a single dose of an antipsychotic drug (data were averaged over all antipsychotics used: clozapine: five mice, aripiprazole: three mice, and haloperidol: three mice). Orange shading indicates duration of the mismatch event. Mean (lines) and 90% confidence interval (shading) are calculated as a hierarchical bootstrap estimate for each time bin (see Methods and *Supplementary file 1*). (**B**) As in A, but for drifting grating onsets.

*et al., 2007*), most signals likely come from the superficial 100 μm of tissue (*Allen et al., 2017*). In the case of the *Tlx3* recordings, this would mean that we were recording a mixture of axonal and dendritic signals of the *Tlx3* neurons that have their soma in L5 but have a dense net of axons and dendrites in L1. In the case of the brain wide expression of GCaMP using an AAV-PHP.eB in C57BL/6 mice, the signals we recorded likely also include contributions from thalamic axons. In addition to this, there is a hemodynamic contribution to the signal (*Allen et al., 2017*; *Ma et al., 2016*; *Valley et al., 2020*) that is the result of varying occlusion by blood vessels. We attempted to correct for the hemodynamic response using a 405 nm wavelength isosbestic illumination of GCaMP6 but could not recover the hemodynamic responses measured in eGFP experiments. Accurate correction likely requires calibration on eGFP measurements (*Valley et al., 2020*). Given that hemodynamic responses should not systematically differ between different mouse lines, and that in the crystal skull preparation the contribution of hemodynamic occlusion to increases in fluorescence was relatively small (*Figure 1—figure supplement 1*), these hemodynamic artifacts should not substantially alter our conclusions. Lastly, it should be kept in mind that responses in these widefield recordings inherently reflect population averages. Assuming there is some form of lateral inhibition among neurons, population averages need not reflect responses of neurons that are tuned to the specific stimulus presented. For example, the suppression to grating stimuli we observed in V1 *Tlx3* positive L5 IT neurons could be the result of a subset of *Tlx3* neurons that are strongly responsive to the specific grating presented suppressing the rest of the population of *Tlx3* neurons. The sum of both could then result in a net decrease in calcium activity. Thus, an understanding of the mechanism that results in opposing population responses of *Tlx3* positive L5 IT neurons to positive and negative prediction errors will require measurements with single neuron resolution.

In our experiments we have focused on *Tlx3* positive L5 IT neurons, but given that L5 IT neurons are densely connected to L5 pyramidal tract neurons (*Anderson et al., 2010*; *Kiritani et al., 2012*) and to L6 neurons (*Xu et al., 2016*; *Yamawaki and Shepherd, 2015*), it is not unlikely that these neuron types would follow similar patterns of activity and exhibit similar susceptibility to the influence of antipsychotic drugs. In the comparison of closed and open loop locomotion onsets, we do indeed find that *Ntsr1* positive L6 excitatory neurons exhibit similar levels of dissimilarity as those observed in *Tlx3* positive L5 IT neurons (*Figure 1L*). It is possible that responses in the *Ntsr1* recordings, however, were noisier than those in the *Tlx3* recordings (*Figure 1—figure supplement 4*), likely because of the lower dendritic and axonal volume of *Ntsr1* positive neurons in L1. Thus, while we have focused on the *Tlx3* positive L5 IT neurons mainly for reasons of experimental convenience, it is possible that both the differential activation in closed and open loop locomotion onsets as well as the susceptibility of long-range correlations to antipsychotic drugs is a general feature of cortical L5 and L6. Another possibility we currently cannot entirely rule out is that the decorrelation effect we observed in *Tlx3* positive L5 IT neurons is specific to the *Tlx3*-Cre mouse line. However, this seems unlikely given that all mouse lines we used were maintained on the same C57BL/6 background.

It is still unclear what the relevant sites of action in the brain are of antipsychotic drugs. Antipsychotic drugs are often antagonists of the D2 receptors, which are most strongly expressed in striatum. Consequently, it is sometimes assumed that the relevant site of action of antipsychotics is the striatum. Consistent with this, an increase in striatal dopamine activates auditory cortex (*Maia and Frank, 2017*) and mediates hallucination-like perceptions in mice (*Schmack et al., 2021*). It is intriguing, however, to speculate that the effects we observed in L5 IT neurons, which provide a dominant input to striatum (*Morita et al., 2019*), may be an alternate site of action. While it is possible that the primary site of action of the antipsychotic drugs with respect to the L5 decorrelation phenotype we observed is striatum, this is unlikely. Changes to striato-thalamo-cortical loops could appear as changes in functional connectivity observed in cortex. However, feedback from striatum to cortex is exclusively mediated through thalamus, and neither L4, the primary thalamo-recipient layer in cortex, nor our brain wide recordings exhibited any of the effects we saw in L5 IT neurons. Thus, the effects we saw in L5 IT neurons are either not inherited from striatum or are mediated by a separate set of thalamic neurons via direct thalamic input to deep cortical layers (*Constantinople and Bruno, 2013*; *Douglas and Martin, 1991*). Thalamocortical collaterals in deep layers, however, come from thalamic neurons that project primarily to superficial layers (*Oberlaender et al., 2012*). Thus, we think it is unlikely that the observed effects in L5 IT neurons were inherited from striatum. Independent of whether the treatment-relevant site of action of antipsychotic drugs is indeed in L5

IT neurons, or whether the observed decorrelation of activity is relevant to antipsychotic potency of these drugs, we speculate that the long-range decorrelation of activity could be used as a basis for a functional screen for compounds with antipsychotic efficacy. Given that in the brain wide recordings the decorrelation of activity was weaker compared to that observed in *Tlx3* positive L5 IT neurons, it might be possible to test for similar effects in human subjects using layer-specific fMRI recordings (*Haarsma et al., 2022*).

A rather surprising implication of our results is that the dominant mode of coupling in cortex might be lateral, not vertical. Two findings would argue for this interpretation. First, while L5 was differentially activated across cortical regions during closed and open loop locomotion onsets, L2/3 exhibited rather limited differences. Second, antipsychotics decorrelated L5 activity but had no effect on L2/3. Both findings could be explained if the long-range lateral coupling within L5 is stronger than the local vertical coupling between L2/3 and L5.

One of the diseases treated with antipsychotic drugs is schizophrenia. It has been speculated that disrupted resting-state functional connectivity in brain networks involving cortex is central to the pathogenesis of schizophrenia (*Li et al., 2017*; *Northoff and Duncan, 2016*). Treatment-naïve schizophrenia patients predominantly exhibit increases in resting-state functional connectivity between cortical regions, while medicated schizophrenia patients exhibit more balanced fractions of increased and decreased functional connections (*Anticevic et al., 2015*; *Du et al., 2021*; *Li et al., 2017*). Auditory hallucinations, one of the hallmark symptoms of schizophrenia, are associated with increased activation of diverse cortical regions (*Catafau et al., 1994*; *Dierks et al., 1999*; *Shergill et al., 2000*). Thus, it is conceivable that antipsychotic drugs counteract this by reducing the strength of long-range cortical connectivity mediated by L5 IT neurons, independent of whether the increased activity in schizophrenic patients is indeed driven by aberrant increases in the strength of long-range cortical connections or is a consequence of an aberrant increase in the strength of striato-thalamo-cortical loops. However, a direct involvement of cortical circuits in the pathogenesis of schizophrenia is consistent with the glutamate hypothesis of schizophrenia. This hypothesis is based on the finding that NMDA receptor antagonists induce schizophrenia-like symptoms (*Moghaddam and Javitt, 2012*), and that disturbances of NMDA and AMPA receptor-related gene expression are associated with schizophrenia (*Harrison and Weinberger, 2005*; *Singh et al., 2020*). In V1, it has been shown that NMDA receptors are necessary during visual development to establish normal prediction error signaling (*Widmer and Keller, 2021*). Thus, it is conceivable that a disruption of glutamatergic signaling in cortex that alters the way internal representations are activated by prediction errors is central to the etiology of schizophrenia. If so, we should find impairments in all domains of cortical function, including sensory processing. In the case of the sensory cortex, cortical dysfunctions should be apparent earlier in development, concurrent with the critical period for plasticity in sensory areas of cortex. While the core symptoms of schizophrenia typically appear in late adolescence, sensory processing is indeed impaired in schizophrenic patients (*Javitt, 2009*) and is apparent, for example, in the reduction of surround suppression in V1 (*Yoon et al., 2009*). Interestingly, some of these disturbances are detectable already during childhood as visuo-perceptual and reading anomalies in some subjects that later develop schizophrenia (*Parellada et al., 2017*). Positive symptoms of schizophrenia can be explained in a predictive processing framework of cortical function as a disturbance of a prediction error-based update of internal representations (*Fletcher and Frith, 2009*; *Frith, 2005*). There are two types of prediction error neurons, positive and negative, and, given that cortex likely implements a non-hierarchical variant of predictive processing (*Garner and Keller, 2022*), two types of internal representation neurons. One set of internal representation neurons is excited by positive prediction errors and inhibited by negative prediction errors, and the other one is inhibited by positive prediction errors and excited by negative prediction errors (*Keller and Mrsic-Flogel, 2018*). *Tlx3* positive L5 IT neurons exhibited opposing responses to positive and negative prediction errors (*Figure 3*), and, while several different interpretations are conceivable, these responses would be consistent with those of internal representation neurons. Given that the effects of antipsychotic drugs are predominantly observed in L5 IT neurons, and that anesthesia is thought to decouple L5 neuron soma from their apical tuft and thereby prevent conscious perception (*Suzuki and Larkum, 2020*), we speculate that *Tlx3* positive L5 IT neurons may be one type of internal representation neuron and one of the functionally relevant sites of antipsychotic action.

# Methods

**Key resources table**

| Reagent type (species) or resource | Designation | Source or reference | Identifiers | Additional information |
|---|---|---|---|---|
| Strain, strain background (adeno-associated virus) | AAV-PHP.eB-hSyn1-jGCaMP7f ($10^{13}$ GC/ml) | FMI vector core | vector.fmi.ch | |
| Strain, strain background (adeno-associated virus) | AAV-PHP.eB-EF1α-GCaMP6s ($10^{13}$ GC/ml) | FMI vector core | vector.fmi.ch | |
| Strain, strain background (adeno-associated virus) | AAV-PHP.eB-DIO-EF1α-GCaMP6s ($10^{13}$ GC/ml) | FMI vector core | vector.fmi.ch | |
| Strain, strain background (adeno-associated virus) | AAV-PHP.eB-DIO-EF1α-jGCaMP7f ($10^{12}$ GC/ml) | FMI vector core | vector.fmi.ch | |
| Strain, strain background (adeno-associated virus) | AAV-PHP.eB-EF1α-eGFP ($10^{13}$ GC/ml) | FMI vector core | vector.fmi.ch | |
| Strain, strain background (adeno-associated virus) | AAV-PHP.eB-EF1α-DIO-eGFP ($10^{15}$ GC/ml) | FMI vector core | vector.fmi.ch | |
| Chemical compound, drug | Fentanyl citrate | Actavis | CAS 990-73-8 | Anesthetic compound |
| Chemical compound, drug | Midazolam (Dormicum) | Roche | CAS 59467-96-8 | Anesthetic compound |
| Chemical compound, drug | Medetomidine (Domitor) | Orion Pharma | CAS 86347-14-0 | Anesthetic compound |
| Chemical compound, drug | Ropivacaine | Presenius Kabi | CAS 132112-35-7 | Analgesic compound |
| Chemical compound, drug | Lidocaine | Bichsel | CAS 137-58-6 | Analgesic compound |
| Chemical compound, drug | Buprenorphine | Reckitt Benckiser Healthcare | CAS 52485-79-7 | Analgesic compound |
| Chemical compound, drug | Ophthalmic gel (Humigel) | Virbac | N/A | |
| Chemical compound, drug | Flumazenil (Anexate) | Roche | CAS 78755-81-4 | Anesthetic antagonist |
| Chemical compound, drug | Atipamezole (Antisedan) | Orion Pharma | CAS 104054-27-5 | Anesthetic antagonist |
| Chemical compound, drug | N-Butyl-2-cyanoacrylate | Braun | CAS 6606-65-1 | Histoacryl |
| Chemical compound, drug | Dental cement | Heraeus Kulzer | CAS 9066-86-8 | |
| Chemical compound, drug | Metacam | Boehringer Ingelheim | CAS 71125-39-8 | Analgesic compound |
| Chemical compound, drug | Clozapine (powder) | Novartis | CAS 5786-21-0 | Antipsychotic drug |
| Chemical compound, drug | Aripiprazole (intramuscular injection solution 7.5 mg/ml) | Otsuka Pharmaceutical | CAS 129722-12-9 | Antipsychotic drug |
| Chemical compound, drug | Haloperidol (intramuscular injection solution 5 mg/ml) | Janssen | CAS 52-86-8 | Antipsychotic drug |
| Chemical compound, drug | Amphetamine (powder) | Hänseler | CAS 60-13-9 | Psychostimulant |
| Chemical compound, drug | Tamoxifen food | Envigo | CAS 10540-29-1 (TD.55125) | Compound to induce Cre expression in CreER mouse lines |
| Genetic reagent *Mus musculus* | *C57BL/6* | Charles River | N/A | |

*Continued on next page*

*Continued*

| Reagent type (species) or resource | Designation | Source or reference | Identifiers | Additional information |
|---|---|---|---|---|
| Genetic reagent *Mus musculus* | *Emx1*<sup>tm1(cre)Krj</sup><br>Alias used here: *Emx1*-Cre | Jackson Laboratories | RRID:IMSR_JAX:005628 | Cre expression in excitatory forebrain neurons |
| Genetic reagent *Mus musculus* | *B6(Cg)-Cux2*<sup>tm3.1(cre/ERT2)Mull</sup>/Mmmh<br>Alias used here: *Cux2*-CreERT2 | MMRRC | RRID:MMRRC_032779-MU | Cre expression in cortical L2/3/4 |
| Genetic reagent *Mus musculus* | *Scnn1a-cre3Aibs/J*<br>Alias used here: *Scnn1a*-Cre | Jackson Laboratories | RRID:IMSR_JAX:009613 | Cre expression in cortical L4 |
| Genetic reagent *Mus musculus* | *Tg(Tlx3-cre)PL56Gsat/Mmucd,*<br>Alias used here: *Tlx3*-Cre | MMRRC | RRID:MMRRC_041158-UCD | Cre expression in cortical L5 IT |
| Genetic reagent *Mus musculus* | *Ntsr1*<sup>GN220Gsat/Mmucd</sup><br>Alias used here: *Ntsr1*-Cre | MMRRC | RRID:MMRRC_017266-UCD | Cre expression in cortical L6 |
| Genetic reagent *Mus musculus* | *Pvalb*<sup>tm1(cre)Arbr</sup><br>Alias used here: *PV*-Cre | Jackson Laboratories | RRID:IMSR_JAX:008069 | Cre expression in PV interneurons |
| Genetic reagent *Mus musculus* | *Vip*<sup>tm1(cre)Zjh</sup><br>Alias used here: *VIP*-Cre | Jackson Laboratories | RRID:IMSR_JAX:010908 | Cre expression in VIP interneurons |
| Genetic reagent *Mus musculus* | *Sst*<sup>tm2.1(cre)Zjh</sup><br>Alias used here: *SST*-Cre | Jackson Laboratories | RRID:IMSR_JAX:018973 | Cre expression in Sst interneurons |
| Genetic reagent *Mus musculus* | *Igs7*<sup>tm148.1(tetO-GCaMP6f,CAG-tTA2)Hze</sup>/J<br>Alias used here: Ai148 | Jackson Laboratories | RRID:IMSR_JAX:030328 | GCaMP6f reporter line |
| Software, algorithm | MATLAB (2021a) | The MathWorks | RRID:SCR_001622 | Data analysis |
| Software, algorithm | LabVIEW | National Instruments | RRID:SCR_014325 | Hardware control |
| Software, algorithm | Python | python.org | RRID:SCR_008394 | Virtual reality |
| Software, algorithm | Panda3D | panda3d.org | N/A | Virtual reality |

## Mice

All animal procedures were approved by and carried out in accordance with the guidelines of the Veterinary Department of the Canton Basel-Stadt, Switzerland (licence no. 2573). The mice used for cell type-specfic targeting and subsequent widefield imaging in this study were kept on a C57BL/6 background and were of the following genotype: 6 C57BL/6 (Charles River Laboratories), 4 *Emx1*<sup>tm1(cre)Krj</sup> (alias: *Emx1*-Cre) to target expression to cortical excitatory neurons (**Gorski et al., 2002**), 4 *B6(Cg)-Cux2*<sup>tm3.1(cre/ERT2)Mull</sup>/Mmmh (alias: *Cux2*-CreERT2) to target expression to L2/3 and L4 excitatory neurons (**Franco et al., 2012**), 7 *Scnn1a-cre3Aibs/J* (alias: *Scnn1a*-Cre) to target expression to a subset of L4 excitatory neurons (**Madisen et al., 2010**), 25 *Tg(Tlx3-cre)PL56Gsat/Mmucd* (alias: *Tlx3*-Cre) to target expression to a subset of L5 IT neurons (**Gerfen et al., 2013**), 3 *Ntsr1*<sup>GN220Gsat/Mmucd</sup> (alias: *Ntsr1*-Cre) to target expression to a subset of L6 excitatory neurons (**Gong et al., 2007**), and, to target inhibitory subpopulations, we used 2 *Pvalb*<sup>tm1(cre)Arbr</sup> (alias: *PV*-Cre) (**Hippenmeyer et al., 2005**), 6 *Vip*<sup>tm1(cre)Zjh</sup> (alias: *VIP*-Cre) (**Taniguchi et al., 2011**), and 5 *Sst*<sup>tm2.1(cre)Zjh</sup> (alias: *SST*-Cre) (**Taniguchi et al., 2011**) mice. *Igs7*<sup>tm148.1(tetO-GCaMP6f,CAG-tTA2)Hze</sup>/J (alias: Ai148) (**Daigle et al., 2018**, Jackson Laboratories, stock #030328) mice were used as breeders to drive GCaMP6f expression in Cre positive neurons. An additional 8 C57BL/6 and 7 *Tlx3*-Cre mice were used in experiments controlling for hemodynamic responses. To induce expression of Cre in the *Cux2*-CreERT2 line, mice were provided for at least 1 week with tamoxifen containing food (Envigo, 400 mg tamoxifen per kg of food) as their sole food source. Mice were group-housed in a vivarium (light/dark cycle: 12/12 hr). Experimental mice were of either sex.

## Surgery and virus injections

For all surgical procedures, mice were anesthetized using a mixture of fentanyl (0.05 mg/kg; Actavis), midazolam (5.0 mg/kg; Dormicum, Roche), and medetomidine (0.5 mg/kg; Domitor, Orion). In a subset of mice (6 C57BL/6, 4 *Emx1*-Cre, and 4 *SST*-Cre mice), we injected an AAV vector based on the PHP.eB capsid (**Chan et al., 2017**) retro-orbitally (6 µl each eye of at least $10^{13}$ GC/ml) to drive expression of GCaMP under either the EF1α or hSyn1 promoter pan-neuronally, or in a cell type-specific

manner using a double-floxed inverted open reading frame (DIO) construct. For hemodynamic response control experiments, we used an AAV-PHP.eB-EF1α-eGFP to pan-neuronally express eGFP (*Figure 1—figure supplement 1A and B*) or an AAV-PHP.eB-EF1α-DIO-eGFP to restrict eGFP expression to L5 IT neurons in *Tlx3*-Cre mice (*Figure 1—figure supplement 1C and D*, *Figure 4—figure supplement 1A*, *Figure 5—figure supplement 1C–E*). To improve optical access to the cortex, we implanted crystal skull cranial windows (*Kim et al., 2016*). Prior to removing the skull plate overlying dorsal cortex, we recorded the location of bregma relative to other landmarks on the skull. Super-glue (Pattex) was used to glue the crystal skull in place in the craniotomy. In a subset of mice, for the comparison of hemodynamic responses (*Figure 1—figure supplement 1A and B*), we used a clear skull preparation (*Guo et al., 2014*). To do this, we attached the crystal skull coverslip directly onto the cleaned skull surface with a three-component polymer (C&B Metabond, Parkell). A custom-machined titanium head bar was attached to the skull using dental cement (Paladur, Heraeus). An epifluorescence overview image was taken to mark reference points on the dorsal cortical surface. Anesthesia was antagonized by an intraperitoneal injection of a mixture of flumazenil (0.5 mg/kg; Anexate, Roche) and atipamezole (2.5 mg/kg; Antisedan, Orion Pharma). For peri- and post-operative analgesia, we injected buprenorphine (0.1 mg/kg; Reckitt Benckiser Healthcare (UK) Ltd.) and Metacam (5 mg/kg; Boehringer Ingelheim). Imaging commenced at the earliest 1 week after head bar implantation or 3 weeks after retro-orbital AAV injection. Clozapine (Novartis), aripiprazole (Otsuka Pharmaceutical), haloperidol (Janssen), amphetamine (Hänseler), or saline (0.9% NaCl in $H_2O$) was intraperitoneally injected at 0.2 µg/g - 10 µg/g, 0.2 µg/g, 0.1 µg/g, 4 µg/g body weight, or undiluted, respectively.

## Virtual reality setup and stimulus design

For all experiments we used a virtual reality setup as previously described (*Leinweber et al., 2014*). Mice were head-fixed and free to run on a spherical, air-supported Styrofoam ball. A virtual corridor was projected (using a Samsung SP-F10M projector) onto a toroidal screen positioned in front of the mouse covering a field of view of approximately 240 degrees horizontally and 100 degrees vertically. Recordings were blocked into 5 min sessions. Experiments typically commenced with four sessions of a closed loop condition in which the rotation of the spherical treadmill was coupled to the movement in a virtual corridor. To introduce mismatches, we broke the coupling between locomotion and visual flow by briefly halting visual flow for 1 s at random times. Closed loop sessions were followed by four sessions of an open loop condition, in which rotation of the spherical treadmill and movement in the virtual corridor were decoupled. The visual stimulus consisted of a replay of the visual flow recorded in the previous closed loop condition. Following this we recorded two sessions of what we refer to as a dark condition in which the virtual reality was switched off. Please note, this was not complete darkness, primarily because the shielding of the blue excitation light was not light proof. Finally, we recorded activity in two sessions of a grating condition in which we presented drifting grating stimuli covering the full toroidal screen placed in front of the mouse (eight directions of 6 s±2 s [mean ± SD] duration each with an inter-grating interval of 4.5 s±1.5 s [mean ± SD] gray screen, presented in randomized order). These sessions, totaling approximately 1 hr of recording time per day, were acquired for 3 consecutive days. For mice in which we went on to test the effect of drugs on dorsal cortex activity, the first post-drug injection data were acquired +1 hr after drug injection (+30 min in the case of amphetamine because of faster onset kinetics of the drug) and further data were collected at the +24 hr and +48 hr timepoints. Data in the manuscript are shown as the combined data over either all data collected before drug injection (labeled as naive) or after substance injection (labeled as clozapine, aripiprazole, haloperidol, amphetamine, or saline, respectively).

## Data acquisition

All widefield imaging experiments were performed on a custom-built macroscope consisting of commercially available objectives mounted face-to-face (Nikon 85 mm/f1.8 sample side, Nikon 50 mm/f1.4 sensor side). We used a 470 nm LED (Thorlabs) powered by a custom-built LED driver for exciting GCaMP (or eGFP) fluorescence through an excitation filter (SP490, Thorlabs) reflected off a dichroic mirror (LP490, Thorlabs) placed in the parfocal plane of the objectives. Green fluorescence was collected through a 525/50 emission filter on an sCMOS camera (PCO edge 4.2). Apertures on objectives were usually kept maximally open and the current at the LED driver was used to adjust fluorescence intensity to a value that was kept below 25% of the maximum dynamic range of

the sensor. In cases where this was not possible (e.g. transfection of eGFP yielded extremely bright fluorescence, often visible with the naked eye), objective apertures were gradually reduced to avoid overexposure of the sensor. LED illumination was adjusted with a collimator (Thorlabs SM2F32-A) to achieve homogenous illumination across the surface of the cranial window. The resulting Gaussian profile of the illumination cone was further trimmed with black tape on the sample side objective to avoid light shining directly into the mouse's eye. An Arduino board (Arduino Mega 2560) was used to control LED onsets synced to the frame trigger signal of the camera. The duty cycle of the 470 nm LED was 90%. Raw images were acquired at full speed (100 Hz; except for eGFP controls, which were acquired at 50 Hz effective frame rate) and full dynamic range (16 bit) of the sensor. The raw images were cropped on-sensor and the resulting data was streamed to disk with custom-written software in LabVIEW (National Instruments), resulting in an effective pixel size of 60 $\mu m^2$ at a standardized imaging resolution of 1108 pixels × 1220 pixels (1.35 MP).

Two-photon calcium imaging was performed using a custom-built microscope based on a Thorlabs B-scope (*Leinweber et al., 2014*). Illumination source was a tunable femtosecond laser (Coherent Chameleon) tuned to 930 nm. Emission light was band-pass filtered using a 525/50 filter for GCaMP and detected using a GaAsP photomultiplier (H7422, Hamamatsu). Photomultiplier signals were amplified (DHPCA-100, Femto), digitized (NI5772, National Instruments) at 800 MHz, and band-pass filtered around 80 MHz using a digital Fourier-transform filter implemented in custom-written software on an FPGA (NI5772, National Instruments). The scanning system of the microscope was based on a 12 kHz resonant scanner (Cambridge Technology). Images were acquired at a resolution of 750 pixels × 400 pixels at a 60 Hz frame rate and a piezoelectric linear actuator (P-726, Physik Instrumente) was used to move the objective (Nikon 16×, 0.8 NA) in steps of 15 µm between frames to acquire images at four different depths. This resulted in an effective frame rate of 15 Hz. The field of view was 350 µm × 350 µm.

## Data processing

Off-line data processing and data analysis (section below) were done with custom-written MATLAB (2021a) scripts. For widefield macroscope imaging, raw movie data was manually registered across days by aligning subsequent mean projections of the data to the first recorded image sequence. We placed ROIs relative to readily identifiable anatomical landmarks that had been previously noted during cranial window surgery (see above). This resulted in the selection of six 20 pixels × 20 pixels ROIs per hemisphere. Of those ROIs, we calculated the activity as the $\Delta F/F_0$, wherein $F_0$ was the median fluorescence of the recording (approximately 30,000 frames in a 5 min recording). $\Delta F/F_0$ was corrected for slow fluorescence drift caused by thermal brightening of the LED using 8th percentile filtering with a 62.5 s moving window similar to what was described previously for two-photon imaging (*Dombeck et al., 2007*).

Raw two-photon imaging data were full-frame registered to correct for brain motion. Neurons were manually selected based on mean and maximum fluorescence images. Raw fluorescence traces were corrected for slow drift in fluorescence using an 8th percentile filtering with a 15 s window (*Dombeck et al., 2007*). $\Delta F/F$ traces were calculated as mean fluorescence in a selected region of every imaging frame and subsequently subtracted and normalized by the median fluorescence.

## Data analysis

Locomotion and visual flow onsets were determined based on threshold crossing of locomotion or visual flow speed. To select only well-isolated onsets, we used a speed threshold of 30 cm/s and excluded all onsets with locomotion or visual flow in the 3 s window preceding the onset. To increase the number of locomotion onsets that passed the speed threshold criterion for the eGFP imaging data (*Figure 1—figure supplement 1*, *Figure 4—figure supplement 1A*), we excluded only those onsets if there was locomotion in the 1 s window preceding onset.

All stimulus response curves (*Figures 1F–L, 3B, C, E, F, 4 and 8*, *Figure 1—figure supplements 1–4*, *Figure 4—figure supplement 1A B, E, and F*) were baseline-subtracted. The baseline subtraction window for unpredictable stimuli (mismatches and grating onsets) was –200 ms to 0 ms before stimulus onset. For closed loop and open loop locomotion onsets, and open loop visual flow onsets, to accommodate calcium indicator offset dynamics and potential anticipatory activity, we placed the baseline subtraction windows for these onsets at –2900 ms to –2700 ms for GCaMP imaging, or –900

ms to –700 ms for eGFP imaging. Varying this window within the bounds of offset dynamics of the indicator and anticipatory activity onset did not change the conclusions drawn in this manuscript. Analyses in which image sequences were aligned to event onsets (*Figures 1C–D and 3*) used the same parameters as above.

On rare occasions, we observed seizure-like activity after antipsychotic drug injection. These events resulted in atypically high levels of correlation, and to avoid them from influencing our conclusions we excluded these data from further analysis. We excluded sessions if they contained an event in which average activity of all 12 ROIs remained elevated above 30% ΔF/F for more than 10 s. This criterion led to the exclusion of 0.26% of data (10 out of 3801 of 5 min recording sessions).

To quantify the similarity of closed loop and open loop locomotion onset responses shown in *Figure 1L*, we calculated the correlation coefficient between the averaged responses (onsets and mice, 1 s moving average) in a window –5 s to +3 s, for each ROI. For *Figure 1—figure supplement 4J*, we first averaged over onsets in either left or right V1, which resulted in an average locomotion onset response curve for left and right V1, for each mouse and condition (closed and open loop). These curves were subsequently used for calculating the correlation coefficient in the same window as above. The correlation coefficients from left and right V1 are then averaged and reported as individual data points in the figure panel. For this analysis, we excluded mice in which the responses were smaller than 1% ΔF/F in either of the two conditions (3 of 54 mice), as the correlation analysis yields noise in the absence of a clear response.

For the correlation analyses shown in *Figures 5–7*, *Figure 5—figure supplement 1*, *Figure 6—figure supplements 1 and 2*, we first calculated the correlation coefficient of neuronal activity (or fluorescence change, in the case of *Tlx3-Cre* mice that had been retro-orbitally injected with an AAV-PHP.eB-Ef1α-DIO-eGFP to express eGFP in L5 IT neurons) for all possible pairs of the 12 ROIs. We then determined the distance between each pair of ROIs and normalized this value by the distance between bregma and lambda obtained from the images during initial surgery for each mouse (*Figure 5—figure supplement 1A*). We plotted the correlation against the distance for each pair of ROIs and binned the data in a 40×40 grid and smoothed the resulting image with a Gaussian filter (*Figure 5—figure supplement 1B*), to obtain the heatmaps shown in *Figures 6A, B, D, E, 7A, B, D, E, G, and H*, *Figure 5—figure supplement 1B–D, G, H*, *Figure 6—figure supplement 1A and B*, *Figure 6—figure supplement 2C and D*. Contour lines in the heatmaps were then drawn to 50% of the maximum value in the plot. The boxplot quantifications (*Figures 6C, F, 7C, F, and I*, *Figure 5—figure supplement 1E and H*, *Figure 6—figure supplement 1C*, *Figure 6—figure supplement 2E*) were calculated as the change in correlation coefficient of activity after and before treatment for all pairs of ROIs, normalized to the value before treatment and split into short- and long-range using a threshold of 0.9 of the bregma-lambda distance for each mouse. For pairwise neuronal correlations that had been recorded with two-photon imaging (*Figure 6—figure supplement 1D*), we calculated the correlation coefficient separately for all pairs of neurons within each field of view. We noticed extreme synchrony of activity of L5 IT soma recorded in 1 field of view with two-photon imaging after clozapine injection and show both distributions for all data as well as the distribution when this 1 field of view is excluded.

For *Figure 1—figure supplement 2*, onset aligned data were first averaged for each mouse and then summed in the given combinations. The SEM was calculated over mice.

To obtain the differences in the correlation between activity and locomotion after and before clozapine injection (*Figure 4—figure supplement 1D*), we first averaged all correlation coefficients in the individual sessions of a condition (closed loop, open loop, or dark) to generate one correlation coefficient per mouse and ROI, for each condition. We then calculated and plotted the difference between these correlation values after and before clozapine injection.

We used a hierarchical bootstrap (*Figure 2C and D*), or a bootstrap (where data were not nested, *Figure 1L* and *Figure 1—figure supplement 4J*), distribution to estimate p-values. For hierarchical bootstrap, we first resampled the data (with replacement) at the level of mice, and then, from the selected mice, resampled for recording sessions. In cases where data were not nested, bootstrap samples were drawn from the distribution of data points. We then computed the mean of this bootstrap sample and repeated this 10,000 times to generate a bootstrap distribution of the mean estimate. The p-value was then defined as the proportion of bootstrap samples higher (or lower, depending on the hypothesis) than zero. For *Figure 1L* and *Figure 1—figure supplement 4J*, to account for multiple

comparisons, we used family-wise error correction. The significance indicators in the figure panels therefore reflect an adjusted significance threshold, $\alpha_{adj}$, according to the formula $\alpha_{adj} = \alpha/m$, wherein $\alpha$ is 0.05, 0.01, or 0.001, and m denotes the number of groups (genotypes, m=9 in both figure panels).

For widefield imaging time course data (*Figures 1F–L, 3B, C, E, F, 4 and 8*, *Figure 1—figure supplement 1*, *Figure 1—figure supplement 3A and C*, *Figure 1—figure supplement 4*, *Figure 4—figure supplement 1A and E*), we used a hierarchical bootstrap estimate of the mean (*Saravanan et al., 2020*) and the 90% confidence interval. This was done by generating 1000 bootstrap samples chosen hierarchically (first over mice, then over onsets) for each time bin (10 ms) in the plot.

To compare the similarity of closed and open loop locomotion onset responses, we used a one-way analysis of variance (ANOVA) with genotype (*Figure 1L*) or mice (*Figure 1—figure supplement 4J*) as factors (*Supplementary file 1*), respectively, followed by a bootstrap distribution-based estimate of p-values as described above.

## Acknowledgements

We thank Tingjia Lu for production of the AAV vectors, Rebecca Jordan for performing the clear skull implants used in the hemodynamic response measurements, and members of the Keller lab for discussion and support. This project has received funding from the Swiss National Science Foundation, the Novartis Research Foundation, and the European Research Council (ERC) under the European Union's Horizon 2020 research and innovation programme (grant agreement No 865617).

## Additional information

### Competing interests

Matthias Heindorf: Matthias Heindorf is author on a patent to use layer 5 decorrelation to screen for antipsychotic efficacy application EP22153051.1. Georg B Keller: Georg Keller is author on a patent to use layer 5 decorrelation to screen for antipsychotic efficacy.

### Funding

| Funder | Grant reference number | Author |
|---|---|---|
| Swiss National Science Foundation | | Georg B Keller |
| European Research Council | 865617 | Georg B Keller |
| Novartis Research Foundation | | Georg B Keller |

The funders had no role in study design, data collection and interpretation, or the decision to submit the work for publication.

### Author contributions

Matthias Heindorf, Conceptualization, Resources, Data curation, Software, Formal analysis, Validation, Investigation, Visualization, Methodology, Writing – original draft, Project administration, Writing – review and editing; Georg B Keller, Conceptualization, Supervision, Funding acquisition, Writing – original draft, Writing – review and editing

### Author ORCIDs

Matthias Heindorf ⓘ http://orcid.org/0000-0002-7866-7601
Georg B Keller ⓘ https://orcid.org/0000-0002-1401-0117

### Ethics

All animal procedures were approved by and carried out in accordance with guidelines of the Veterinary Department of the Canton Basel-Stadt, Switzerland (licence number 2573).

Reviewer #1 (Public review): https://doi.org/10.7554/eLife.86805.4.sa1

Author response https://doi.org/10.7554/eLife.86805.4.sa2

## Additional files

### Supplementary files
• Supplementary file 1. A list of all statistical comparisons for bar and violin plots. Note, for *Figure 1L* and *Figure 1—figure supplement 4J*, the significance thresholds were corrected for multiple comparisons (see Methods).

• Supplementary file 2. A list of number of mice used for each figure and the corresponding indicator expression strategy.

• MDAR checklist

### Data availability
All data and code to reproduce the findings from the paper are freely available for download at https://doi.org/10.5281/zenodo.10783459. The lab's core analysis code as well as all code used for data acquisition are available at https://sourceforge.net/projects/iris-scanning/ (*Keller and Widmer, 2024*).

The following dataset was generated:

| Author(s) | Year | Dataset title | Dataset URL | Database and Identifier |
|---|---|---|---|---|
| Keller GB, Heindorf M | 2024 | Antipsychotic drugs selectively decorrelate long-range interactions in deep cortical layers | https://doi.org/10.5281/zenodo.10783459 | Zenodo, 10.5281/zenodo.10783459 |

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
