## [Editor Report · eLife assessment]

This **important** study uses calcium imaging in mice to advance our understanding of the effect of antipsychotic drugs on neural functioning. The evidence supporting the conclusions is **convincing**, and this work will be of interest to neuroscientists working on visual processing and psychosis researchers.

---

## [Referee Report · Reviewer #1 (Public review)]

The authors present a study of visuo-motor coupling primarily using wide-field calcium imaging to measure activity across the dorsal visual cortex. They used different mouse lines or systemically injected viral vectors to allow imaging of calcium activity from specific cell-types with a particular focus on a mouse-line that expresses GCaMP in layer 5 IT (intratelencephalic) neurons. They examined the question of how the neural response to predictable visual input, as a consequence of self-motion, differed from responses to unpredictable input. They identify layer 5 IT cells as having a different response pattern to other cell-types/layers in that they show differences in their response to closed-loop (i.e. predictable) vs open-loop (i.e. unpredictable) stimulation whereas other cell-types showed similar activity patterns between these two conditions. They also analyzed the responses to visuomotor prediction errors obtained by briefly pausing the display while the mouse is running, causing a negative prediction error, or by presenting an unpredicted visual input causing a positive prediction error. Surprisingly, they find that presentation of a visual grating actually decreases the responses of L5 IT cells in V1. They interpret their results within a predictive coding framework that the last author has previously proposed. The response pattern of the L5 IT cells leads them to propose that these cells may act as 'internal representation' neurons that carry a representation of the brain's model of its environment. Though this is rather speculative. They subsequently examine the responses of these cells to anti-psychotic drugs (e.g. clozapine) with the reasoning that a leading theory of schizophrenia is a disturbance of the brain's internal model and/or a failure to correctly predict the sensory consequences of self-movement. They find that anti-psychotic drugs strongly enhance responses of L5 IT cells to locomotion while having little effect on other cell-types. Finally, they suggest that anti-psychotics reduce long-range correlations between (predominantly) L5 cells and reduce the propagation of prediction errors to higher visual areas and suggest this may be a mechanism by which these drugs reduce hallucinations/psychosis.

This is a large study containing a screening of many mouse-lines/expression profiles using wide-field calcium imaging. Wide-field imaging has its caveats, including a broad point-spread function of the signal and susceptibility to hemodynamic artifacts, which can make the interpretation of results difficult. The authors acknowledge these problems and directly address the hemodynamic occlusion problem. It was reassuring to see supplementary 2-photon imaging of soma to complement this data-set, even though this is rather briefly described in the paper. Some comparisons in the paper are underpowered as a result of including only a small number of mice (e.g. the PV, Ntsr1 and Cux2 mice) and results involving these mice should be cautiously interpreted, but in general the results are robust. Overall the paper's strengths are its identification of a very different response profile in the L5 IT cells compared to other layers/cell-types which suggests an important role for these cells in handling integration of self-motion generated sensory predictions with sensory input. The interpretation of the responses to anti-psychotic drugs is more speculative but the result appears robust and provides an interesting basis for further studies of this effect with more specific recording techniques and possibly behavioral measures.

---

## [Author Response]

The following is the authors’ response to the previous reviews.

**Reviewer #1 (Recommendations For The Authors):**
Revised manuscriptThe authors have addressed most of my points, but I still have one outstanding concern about the statistics:My Original Question:I have a few concerns and questions that I would like to see addressed: (1) Figure 1L - the statistics are a little unusual here as the errors are across visual areas rather than across mice or hemispheres. This isn't ideal as ideally, we want to generalize the results across animals, not areas, and the results seem to be driven mostly by V1/RSC. I would like to see comparisons using mice as the statistical unit either in an ANOVA with areas as factors or post-hoc comparisons per area.Author Reply:Based on the assumption that visual cortex should respond to visual stimuli, we would have expected to find a difference between closed and open loop locomotion onset responses in all cell types in visual areas of cortex (a closed loop locomotion onset being the combination of locomotion and visual flow onset, while an open loop locomotion onset lacks the visual flow component). Thus, the first surprise was that in most cell types we found very little difference between these two locomotion onset types. Conversely, in Tlx3-positive L5 IT neurons the difference was apparent well outside of the visual areas of cortex (even though the difference was indeed strongest in V1/RSC). To quantify the extent to which closed and open loop locomotion onsets result in different activity patterns across dorsal cortex we performed the analyses shown in Figures 1L and 2. To make the point that the effect was observable on average across cortical areas, we used cortical area as a unit in Figure 1L. We have added the analysis shown in Figure 1L with mice as the statistical unit as Figure S4J and have added the ANOVA information to Table S1, as suggested.My revised question:The authors have only partially addressed my concerns here. I disagree with the authors that they were making a point about the effect being observable across visual areas. The primary statistical statement they are trying to make is that the similarity between open and closed-loop stimulation is different for Tlx mice, e.g. Line 122: "However, comparing locomotion onsets in mice that expressed GCaMP6 only in Tlx3 positive L5 IT neurons, we found that the activation pattern was strikingly different between closed and open loop conditions" and Line 172-3: "Thus, excitatory neurons of deep cortical layers exhibited the strongest differences between closed and open loop locomotion related activation". These statements are not correctly supported by the statistical analysis as presented in Figure 1L as it is the variability across mice that is relevant to draw this conclusion.

In the example "However, comparing locomotion onsets in mice that expressed GCaMP6 only in Tlx3 positive L5 IT neurons, we found that the activation pattern was strikingly different between closed and open loop conditions (Figure 1D)" we talk about the example mouse shown. We have not changed phrasing here.

We have, however, changed the way we talk about Figure 1L and S4J (the second example given by the reviewer), and have rephrased much of this paragraph. Please note, we have also changed Figure S4J to quantify the difference only for V1.

This is partially addressed by Figure S4J where the authors show standard-errors across mice and report statistics across mice. In Table S1 the statistical test is reported to be a bootstrap test with mice as the statistical unit, however, according to line 985 this was a non-hierarchical bootstrap test. Does this mean that the authors resampled onsets without regard to which mouse they came from to regenerate the response-curves and recalculate the correlation coefficient? Or did they directly resample from the distribution of correlation coefficient values? I suspect the latter, but for some comparisons (e.g. Tlx3 vs PV) there are only two mice in one group, yielding two correlation coefficients, and resampling 2 values 10,000 times would lead to very biased statistics. Either way the approach is far from ideal. There is also no protection against multiple-comparisons in these tests.

We have adapted Figure S4J to include only V1, where we find the largest effect (the text is adapted to reflect this) and have added individual data points as suggested in the following comment. The reviewer is correct that we created a bootstrap distribution by resampling correlation values. This means we are resampling 2, 3, 4, 6, 7, or 14 values depending on the comparison. This should now be clearer in the text. We agree that this is not ideal, but when using mice as a statistical unit, analysis is almost always underpowered. To the best of our knowledge, bootstrap resampling is the best approach to alleviate this problem. Regarding the concern for multiple comparisons: We have now adjusted the significance threshold in Figures 1L and S4J by dividing through the number of groups (here: 9 genotypes).

The ANOVA reported in Table S1 for Figure S4J isn't described in the methods so I can't say what they did and it doesn't seem to be referred to in the text and is non-significant in any case. Figure S4J also only shows summary statistics whereas individual mice should be plotted. The correct statistical test is either a one-way ANOVA with one factor (genotype) with post-hoc tests between the Tlx3 genotype and the others with suitable multiple-comparisons corrections (this may be the non-significant test in table S1). Alternatively, a linear mixed effects model with Genotype as a fixed effect and Mouse as a random intercept term. This approach is more powerful as it would allow them to use data from all locomotion onsets, but it may struggle to fit datasets with only 2 members for certain genotypes. If they wish to make the more extended point that the pattern across visual areas differs between Tlx3 and other mice they could include 'Area' as another (fixed) factor in the design and look for an interaction with Genotype.

The ANOVA was indeed a one-way ANOVA with one factor. We have added this information to the methods. As suggested, we have added individual data points to Figure S4J.

I also agree with the other reviewers that the presentation of standard-errors in Figures 1F-K and elsewhere is somewhat misleading as these are s.e.m. across onsets without taking into account the hierarchical nature of the data. Across mice s.e.m. would give a more accurate view of the variability in the data across the population. I also understand that first averaging across onsets within mice before taking a grand-average throws away a lot of data and s.e.m.s will be considerably larger. The authors should consider linear mixed effects models as an optimal solution for estimating s.e.m. If this is not feasible then the authors could consider showing data from individual mice in a supplementary figure or at least reporting the number of onsets that came from each mouse.

We have now changed all plots in which we show time course data of widefield calcium imaging to show a hierarchical bootstrap estimate of mean and 90% confidence interval of the mean estimate.

**Reviewer #2 (Recommendations For The Authors):**
Congratulations to the authors on the revision! The revised article has substantially improved, and I have no further comments. I am particularly reassured by the new hierarchical bootstrap analyses as well as by the new analysis with mouse as a statistical unit that reproduces the key finding from the analyses with region as a statistical unit. Moreover, the authors added a vehicle control condition which does not yield any results. Therefore, I have no further methodological concerns and removed my mention of this previous weakness from my public review. Also, the readability of the manuscript has much improved in the revised version. Congratulations again on this important work!

We thank the reviewer for the help in improving the manuscript.

**Reviewer #3 (Recommendations For The Authors):**
Comments on rebuttal:(1) It is greatly appreciated that the authors have improved aspects of their statistics, I have revised my comments accordingly.

We are happy to hear.

(2) However, I should clarify my comments regarding statistical concerns were not merely pertaining to a given Figure (e.g. Figure 1) I was only using it as an example. The authors have redone aspects of their analysis using N = number of mice (for statistics/trace figures), but is there a reason they cannot do this for other problematic figures/traces in the manuscript?

Prompted also by reviewer 1, we have changed all time course plots in the manuscript to show a hierarchical bootstrap estimate of mean and 90% confidence interval of mean.

Using mice as a statistical unit throughout the manuscript unfortunately is not viable in most cases, as we simply do not have enough mice in our dataset and statistical tests based on mice would be underpowered. The manuscript currently contains data from 77 mice, and we would likely need multiples of that to do statistics over mice.

For Figure 1 - I do take the point why regions are being used as the independent N (though the authors justification should be made more clearly in the manuscript) making an N of 12 (though I am less clear why the same region across 2 hemispheres is counted as 2 Ns instead of 1; are they really independent?). However, I am less clear as to the choice in N in other figures. Could the authors clarify this more explicitly in the manuscript.

We use regions as a statistical unit in Figures 6 and 7, S6-S8. Regarding the independence of hemispheres, this depends on cell type and region. E.g. activity in left V1 exhibits a higher correlation with activity left V2am than with right V2 (see Figure 5). On average callosal pairs exhibit correlation levels comparable to near cortical neighbors. See also, other work on the topic, for example (Calhoun et al., 2023).

Regarding choice of N in other figures, this is either “recording session” or “pairs of regions”. We have made this clearer in the figure legends. In the case of testing using recording sessions, the idea is that each recording session constitutes a measurement. Measurements in the same mouse are not independent, and hence we use hierarchical bootstrap for all testing on recording sessions. The choice of “pairs of regions” for the correlation analysis follows from the use of regions as a statistical unit.

(3) Regarding using N = locomotion onsets (or other definitions other than N = mice) when deriving trace averages/SEMs (for example, as in Figure 1) is visually misleading for the reader as it masks the true variability of the data, and even more misleading given that the authors do necessarily use that definition of N in their statistical tests associated with the data (as the authors commented). Whilst the authors have shown some traces with N=mice for some data, is there a reason they cannot do this for all figures in the manuscript? At the very least the practice of using other definitions of N for the purpose of showing trace averages/SEMs should be justified in the MS.

We have replaced all time course plots that used SEM over events (for example locomotion onsets or visual stimuli) with a hierarchical bootstrap estimate of mean and 90% confidence interval of the mean throughout the manuscript. See also response to comment 2 above, and to reviewer 1, comment 4.

References

Calhoun, G., Chen, C.-T., Kanold, P.O., 2023. Bilateral widefield calcium imaging reveals circuit asymmetries and lateralized functional activation of the mouse auditory cortex. Proc. Natl. Acad. Sci. U. S. A. 120, e2219340120. https://doi.org/10.1073/pnas.2219340120